

# Causes and systematics of inundations of the Krasnodar territory on the Russian Black Sea coast

**N. I. Alexeevsky[1, +], D. V. Magritsky[1], K. P. Koltermann[1], I. N. Krylenko[1, 2] and**

**P. A. Toropov[1, 3]**

[1]{Lomonosov Moscow State University, Moscow, Russia}

[2]{Water Problem Institute, Russian Academy of Sciences, Moscow, Russia}

[3]{Institute of Geography, Russian Academy of Sciences, Moscow, Russia}

[+]{deceased in 2015}

Correspondence to: D. V. Magritsky (magdima@yandex.ru)

## Abstract

We analyse inundation situations on the Black Sea coast of Krasnodar territory for the period from 1945 until 2013 and describe the main types of inundations at the coast. Synoptic factors of the formation of extreme rainfalls and rainfall floods, features and regularities of the downstream flood wave transformation in the rivers are studied. Assessments of seasonal and maximum flow of the Black Sea Coast rivers for the period for which hydrometric measurements describe regularities of change of the occurrence of inundations and their characteristics on the coastal terrain, within a year and on perennial time scale.

Most catastrophic and exceptional inundations arise in the summer and in early autumn. Small inundations during the remaining year reflect the seasonal distribution of river flow and floods in the Black Sea Rivers. Extensive and sometimes extreme precipitation dominates the river flow regimes.





The seasonal distribution of small and moderately dangerous inundations reflects, on average, a water regime of two groups of rivers of the coast – to the north of the Tuapse River, and to the south. To the north of the Tuapse River, floods prevail from November until March (to 70 %). They result from precipitation and winter snowmelt during frequent thaw periods. High waters in the cold season of the year often overlap with each other, forming a multipeak high water with 2–3 weeks in duration. In the summer and in early autumn a steady low flow is observed. The total amount of runoff increases both in a southeast direction, and with the altitude of the river basins. Inter-annual variability of mean annual runoff, as well as maximum runoff, on the contrary decreases in the southern direction and with an increasing area of a river basin. The coastal high waters of the rivers of the Sochi part are typical at any time of the year, but more often floods in the cold season result from incessant rains, and thawing snow. Annually up to 25 floods are observed. The principal reason of such distribution is the increase of extreme rainfalls in the warm season.

Orographic features of the coast and detailed features of rainfall cover only a small number of local river basins and a limited area. The geographical correlation of individual rainfall and subsequent floods ceases to be statistically significant for the distances over 40–60 km

The annual flow cycle is mainly determined by strong winter and spring, and weak summer flows: Despite a characteristic distribution of floods and of water flow within a year, almost 71 % of all catastrophic and exceptional inundations took place in July - August (71 %) and in October - November (29 %). The characteristic features of dangerous floods are their rapid formation and propagation, a significant increase of water level (up to 5–7 m and more) and multiple increase of water discharges.

An appreciable increase of the number of inundations in the period from the early 1970s until the early years of the 21[th] century was noted.

Quantitative assessments of risk, hazard and damage for the population and economic activities from accidental inundations in the valleys of the Black Sea coast rivers show that economic and social losses from inundations at the Black Sea coast of the Krasnodar territory are one of the highest in the Russian



Federation. The basic conclusion from recent inundations is the need to consider not only the lower reaches and mouths of the Black Sea coast rivers where the main part of the social and economic development of the coast is concentrated, but also total river basins and catchments. Further, we provide an analysis of the efficiency of the measures applied at the coast to fight inundations and their after-

effect.

# 1   Introduction

Coasts, valleys and river mouths are often subject to the influence of various dangerous hydrological phenomena. Of all dangerous phenomena, inundations result in the most significant economic and

ecological damages, and are the greatest danger to the population. The Black Sea coast of the Krasnodar territory is the most prone to this kind in Russia. In this rather small area, there were five catastrophic inundations during the last 10–20 years, which resulted in huge material damages and considerable human loss. Further nine high and a number of smaller inundations took place. Recent devastating inundations have happened in July and August 2012. Inundations on July 6-7, 2012 have effected both

the coast and the northern slopes of the Caucasus (Krymsk city), and have led to the death of about 170 people and to a material damage of ca. 625 million U.S. dollars.

In general, there is a certain increase in the number and intensification of the magnitude of inundations. Which factors contribute to this impression? Is it the reaction to global and regional climate changes, the intensification of instabilities of the climate system, or the effectiveness of the existing system of

forecasting inundations and flood prevention and the threat the floods pose to the settlements at the coast? What could be the operative prevention measures to minimise the potential damage, and which potential means are available to strengthen this system?

Unfortunately, for too many of these questions there are presently no clear answers, for several principal reasons. First, despite the known significant hydrological hazards, a complex survey of inundations at

the Black Sea coast of the Russian Federation practically is presently not available. Many available





publications have a narrative character and focus on the description of recent catastrophic inundations and their effects (Vorobyev, 2003; Evsjukov et al., 2013; Ermachkov, 2010; Taratunin, 2000; Tkachenko, 2013). Important elements of a scientific analysis are found in some works (Atlas, 2007; Barinov, 2009; Beljakova et al., 2013; Volosuhin, Tkachenko, 2013; Kononova, 2012; Magritsky et al., 2013; Dangerous , 1983; Panov et al., 2012; Sergin et al., 2001; Tkachenko, 2012), and in papers of the Kuban State Agriculture University. The second reason is the lack of high-quality data on inundations observed in the past, especially in the 20th century: the basic characteristics of inundations (areas submerged, their spatial extent, the intensity of their development and duration, of flooding height levels, water levels and discharges), and information on the magnitude and the structure of the estimated damage. The third reason is the continuing sparseness of necessary details and reliable long series of observational data. Hydrological monitoring is not available for all rivers of the coast. The frequency of observations is normally two times a day, and in official hydrological directories the data are published in general averaged over 1 day, whereas high waters form and pass within some hours. Hydrological gauge stations have only a short record length. Many stations have been discontinued; practically there are no rivers with several gauging stations along their course. Moreover, in view of the complex orography and particular features of precipitation, the observations at meteorological stations are not always representative, even for adjacent river basins. Meteorological radars at the coast until recently were not available.

Probably the lack of necessary data, of detailed research, of a better understanding of factors and fundamental physics of the flood phenomenon, and knowledge of their key parameters is one of the reasons for the insufficient efficacy of measures applied at the coast to cope with inundations and their effects. The data collected by the authors and results of our long-term research of the inundation problem, quantitative, graphic and cartographical interpretations of available information allows now to fill the gap in knowledge, and to stimulate new research.



## 2    Objectives of research

The Black Sea coast of the Krasnodar territory of the Russian Federation includes the Temryuk and Tuapse administrative areas (districts), and the cities of Novorossiysk, Anapa, Gelendzhik and Sochi (Fig. 1). These cities also are the main administrative districts by their area, proportion of the developed

and unsettled territories. The total area of the coast is nearly 8015 км$^2$. As a relatively narrow strip of land (its average width is 23 km) the coast extends over 350 km (from the Kerch Strait to the Psou River). The land border of the Black Sea coast coincides with the watershed line between the basins of the Azov and Black Seas.

The coastal terrain is well developed. Over 1.1 million people live here constantly. About 90 % of the

resident population is concentrated in a narrow strip with a width from 0.5 to 8 km and 80 % live in cities and urban settlements. Big cities are Anapa, Novorossiysk, Gelendzhik, Tuapse, Sochi and Adler. It is the largest recreational region of Russia and a fast developing cluster of various sports (in 2014 the XXIIth Olympic winter games took place in Sochi), and a large new business and cultural centre. The number of tourists per year in the region is about 7 million; most part of tourists is concentrated in the

municipal districts along the seacoast.

This region is an important agricultural area of Russia, with a large centre of petroleum refining, production of building materials, transfer of dry and liquid goods, and transportation of natural gas and oil products. There are the important seaports the Gelezny Rog, Novorossiysk and Tuapse.

The Black Sea coast is not homogeneous by its constitution, composition and environment (Panov et al.,

2012; Resources, 1969; Sergin et al., 2001). These distinctions define the specificity of the dangerous natural phenomena in the coastal areas in their kind and hierarchy, and in their impact regarding their development, real or potential damages. A contrasting topographical relief and geological constitution, the irregular distribution of atmospheric precipitation create their heterogeneity. Under hydrological aspects, the Black Sea coast is an isolated basin compounded by numerous basins of small rivers. From

Krasnodar territory 252 watercourses flow into the Black Sea, only 16 % have a length of more than 10



km (Hydrological Survey, 1964). Only three rivers (the Shakhe, Mzymta and Psou) have a length of more than 50 km and a drainage area above 400 км$^2$ (Fig. 1). The drainage density increases in the southeast direction – from 0.3 – 0.5 (and less) to 1 km/km$^2$. A large water slope characterizes almost all the streams. On separate reaches, they look like mountain streams with waterfalls. The floodplain is intermittent and narrow, and usually not developed in the upper reaches and in gorges. On the seaside part, numerous alluvial cones occupy the bottom of river valleys.

Total water resources of the rivers of the Black Sea coast are 7.0–7.5 km$^3$/year, or about 2 % of the total river flow into the Black Sea. The total amount of runoff increases both in a southeast direction, and with the altitude of the river basins. Inter-annual variability of mean annual runoff, as well as maximum runoff, on the contrary decreases in the southern direction and with increasing area of a river basin (table 1, table 2). Reductions of runoff due to economic activities are largest on the rivers of the cities of Anapa and Novorossiysk, i.e. in arid and foothill watersheds. Within these terrains, there are many ponds intercepting a part of river water, and agricultural areas demanding an artificial irrigation. Significant water use practices take place also in the basin of the river Mzymta, mainly hydroelectric engineering, and agricultural, industrial and municipal water consumption. The majority of settlements from Novorossiysk to Sochi is supplied with water, pumped from the thick alluvial depositions under the river channels.

Extensive and sometimes extreme precipitation dominates the river flow regimes. Therefore, the maximum water levels and discharges are observed in any month of a year. To the north from the Tuapse, river floods prevail from November until March (to 70 %). They result from precipitation and winter snowmelt during frequent thaw periods. High waters in the cold season of the year often overlap with each other, forming a multipeak high water with 2–3 weeks in duration. In the summer and in early autumn a steady low flow is observed (Fig. 1). During this period, even rather large rivers can dry up on separate reaches for several days and even several months. Occasionally the low flow is interrupted by high waters caused by heavy rain. In total, 10 – 13 floods per year happen on average. The annual flow



cycle is mainly determined by strong winter and spring, and weak summer flows: for winter and spring, we find about 82 – 86 % of annual runoff, and at the Tuapse River – 75 % (table 1, Fig. 1).

 The coastal high waters of the rivers of the Sochi part are typical at any time of the year, but more often floods in the cold season result from incessant rains, and thawing snow. The rivers Shakhe, Sochi and Psou, basins of which have significant areas and altitudes, demonstrate a similarity of spring high water; but only the Mzymta River has a well-distinguished spring-and-summer high water – from March until August (Fig. 1). It is formed by the melting of the seasonal snow cover, the permanent high-mountain snow cover and of snow-patches, glaciers and rain. The duration of the low flow period is shorter compared to the rivers to the north of the Tuapse River, and in general, the river discharges are higher in this period because they are frequently interrupted by floods. Annually up to 25 floods are observed (on the average 16–20). Moving from the northern to the southern borders of the Sochi part of the coast (table 1) the percentage of the winter-spring season river flow decreases from 75 to 55 – 65 %.

Floods are typical also for small, in fact, temporary water streams (the native name "cleft"). Their surface flow rises only in the season of rainfalls and (or) snowmelt. Absence of water in the channels of such riverbeds during the main part of the year generates the deceptive opinion about hydrological safety at the end of their valleys. Therefore, adverse consequences of inundations here quite often acquire the features of catastrophic events.

## 3    Hydrological data and methods of research

Long-term observation data at 24 hydrological gauging stations of the Federal Hydrometeorology and Environmental Monitoring Service (table 1) form the data basis for our research. The data are presented as averages (for days, 10-day, monthly and annual averages) water levels and water discharges, and by instantaneous maximum and minimum water discharges and levels. Secondly, numerous documentary data on inundations are collected. These data and information are part of the database «Inundations in the river mouths of the European part of Russia» described in (Alekseevsky et al., 2013) which is



available on the web site of the Natural Risk Assessment Laboratory NRAL of the Moscow State University (http://www.nral.org). Thirdly, we used critical marks of the height and rise of water levels which, when exceeded, lead to flooding of flood plains. They are classified as unfavourable (UP) and dangerous (DP) for the population and economic activities. They are particular to each separate reach of

a channel. Fourthly, we took into account daily precipitation data at six meteorological stations (Sochi, Krasnaya Polyana, Tuapse, Dzhubga, Gelendzhik and Anapa) for the period 1945 - 2013 along with regional criteria of dangerous precipitation. According to these criteria, rainfalls in the Tuapse and Sochi districts with an intensity of not less than 50 mm during no more than 1 hour (http://www.yugmeteo.donpac.ru/oj.jsp) are considered as heavy. Rainfalls of not less than 80 mm on

the part of the coast from Anapa to the settlement Dzhubga, 100 mm for the Tuapse district and 120 mm for Sochi during no more than 12 hours are considered very heavy. In mountain areas, the lower limit of very heavy rainfall values is reduced to 50 mm (Tuapse district) and 80 mm (Sochi).

Additionally we used: 1) field data collected by the research projects of the Geographical Faculty of Moscow State University (the authors of the present paper) from 2011 to 2014; 2) the data of

continuous monitoring (time interval of 1 - 10 minutes) of water level at 53 gauges of the automated system of water level monitoring of the flood control system on the rivers of Krasnodar territory since 2012 (http://test.emercit.com/overall.html); 3) actual and archived data of various departments and organizations,  large - scale maps and charts, and data from Internet sources.

These data confirm past inundations at the Black Sea coast of the Russian Federation. The comparison

of the documented events confirmed inundations to maximum water levels ( $H_{max}$ ) and peak discharges ( $Q_{max}$ ) at the gauges, with critical high-rise marks − $H_{floodplain}$, $H_{UP}$, $H_{DP}$, together with daily totals of precipitation at meteorological stations; critical values (or ranges of values) for these hydro-meteorological characteristics  have been defined. Based on these results and long-term series (for the period 1945-2013) of hydro-meteorological characteristics, all probable cases of river inundations in the

past have been determined. The newly documented background, which has not been used in research before, allows us to investigate extend and causes of inundations in this area.



Part of the statistical analysis (verification of significance of linear trends using parametric and nonparametric tests for the significance level $\alpha = 5\%$, the spectral analysis of the data time series, studying the regression relationship between the magnitude of the inundation and the scale of material damage, etc.) has been performed with the program STATISTICA10 (company StatSoft).

5 The collected materials and their analysis have allowed to classify, first, inundations on the Black Sea coast of Krasnodar territory by their causes, and, second, to adapt for the territory under consideration and their rivers existing (in Russia) classifications of inundations by magnitude and after-effects.

## 4    Inundations and their types

10 Inundation in the Russian Federation is perceived as flooding by water of an area adjoining the river or a water body, which leads to a material damage, loss of health of the population or to loss of human lives (Nezhihovsky, 1988). More expanded and with an ecological interpretation of this concept (Dobrovolsky, Istomina, 2006) suggest: «inundation is a temporary flooding of terrain mastered by the human for various purposes, generating negative consequences of social and economic and ecological character expressed in a material and non-material damage». On the contrary, flooding by water of not mastered terrains, not accompanied by a damage, is considered as the natural hydrological process accompanying one of the standard phases of a water regime of a river – spring high water or a flood. It is not considered as inundation.

Taking the formation processes and following the new classification, stated in (Alekseevsky and Magritsky, 2013), at the coast, there are some generic types of inundations, of natural origin. River-flow inundations dominate. At the coast they are generated by high rainfall floods (i.e. during peak discharges and without backwater effect), which sometimes are transformed into in mudflow type. More infrequent, they are induced by an intensive snowmelt in the drainage basins (including the contribution of rain), breakages of dams of ponds and of dammed or glacial lakes. The flood plains of river valleys and river mouths with alluvial cones are also subject to flooding by river waters.



Inundations of the mixed type (№1) – river-flow and rainfall origin – are next by their occurrence. They are also called pluvial inundations. In general, rainfall inundations (a subtype of local meteorological inundations) which also are frequent at the coast are caused by heavy rainfall over the developed areas and by the "inability" of the terrain to quickly absorb or drain rainwater into surface and underground water bodies. The magnitude of rainfall inundations increases if storm drains are functioning badly, therefore in the obvious and dangerous form they happen in settlements, and their frequency increases with the increase of the area of the urbanized terrains. They are also named urban inundations. During pluvial inundations, flooding is caused by river and rainfall waters, and by powerful overland streams formed by rainfall and by waters of "revived" temporary watercourses. These inundations affect not only flood plains, but also river terraces, and sides of the river valley.

The third type are the inundations due to storm surges and wind-wave surges, or inundations again of mixed type (№2) – a flood in a river coincides with a storm surge at the coast, i.e. in conditions of a back-water effect from a sea. These inundations are possible in river mouths at the coast. They are part of a group of coastal inundations.

Other types of inundations (due to ice-jams, tsunami) represent only a potentially small danger to the city of Anapa (Magritsky et al., 2013).

Inundations of one generic type differ by their characteristics (frequency, the area of flooding and the number of the river basins affected by inflow, height and duration of flooding, etc.), and also by size and structure of social and economic damage. The authors do not consider elaborating new approaches concerning such a division. Classifications already existing in Russia (Dobrovolsky and Istomina, 2006; Dobroumov and Tumanovsky, 2002; Malik, 2003; Nezhihovsky, 1988; Taratutin et al., 2011) have been used, including those used by the Ministry of Civil Defence and Emergency Situations of the Russian Federation. Accordingly, the river-flow and mixed type 1 inundations at the Black Sea coast of Krasnodar territory – the subject of this study – can be divided into small (I), moderately dangerous (II), big (III), catastrophic (IV) and exceptional (V). This classification is based on various qualitative and quantitative criteria ranging from frequency, value of excess of $H_{max}$ over critical high-rise marks, the



area of the terrain and number of the settlements (or basins) covered by the influence of inundations and their causes, or the amount of a direct material damage (as a rule at an approximate assessment) and threat for life. Among other criteria, we considered: 1) character of direct damage to industrial objects and road infrastructure, residential buildings, 2) the area and structure of flooding of the mastered

terrain, 3) degree of infringement of way of life and industrial activity of people, 4) necessity of evacuation of people, 5) deterioration of ecological conditions. Unfortunately, as it often happens, not for all the events comparable data are available. In general, for a separately taken river on the Black Sea coast the big inundations occur with a frequency $Q_{max}$ on the average of $4 - 5$ %, catastrophic and exceptional $<2 - 2.5$ % respectively. Long-term observation data at 24 hydrological gauging stations of

the Federal Hydrometeorology and Environmental Monitoring Service.

## 5  Synoptic conditions of the formation of high floods

High floods leading to river-flow inundations, and also flooding by rainfall waters and by overland streams, are formed by the availability of large volumes of water in form of abundant and steady

precipitation, by storm rain and, as a special case, as a result of the destruction of waterspouts over the land. Most catastrophic inundations are a consequence of mesoscale atmospheric processes arising in typical synoptic conditions that form especially powerful overcast. For the Black Sea coast of the Krasnodar territory, the formation of abnormal precipitation results from the topographical relief forcing the air upward and, hence, the process of cloud formation and precipitation.

To analyse the dynamics the atmospheric circulation one can use circulation indices. Among the most well-known are the NAO-index, the Atlantic / West Russia (EA / WR-index), and the Nordic index (SCAND-index). These indices are useful in the analysis of large-scale circulation pattern using results from numerical modelling. For a detailed analysis of the synoptic situation, these indexes are often not representative. We find it more appropriate to use the classic synoptic classifications, based on a

detailed description of synoptic processes of the study area. In Russia, one of the most popular is



Dzerdzeevsky's classification (Dzerdzeevsky, 1975; Kononova, 2012). It uses the concept of "elementary circulation mechanisms," and all synoptic processes combine the four basic types. 1 - zonal circulation, 2 - zonal and meridional circulation, 3 - meridional North, 4 - meridional South. All synoptic processes can be attributed to one of these groups. We use this approach in this study for the detection of the synoptic processes causing strong flooding. We therefore consider those synoptic circulation pattern to cause floods in the Black Sea coast.

1. Mediterranean cyclones generated at the polar front advect abundant rainfall. The greatest recurrence and intensity of these processes is observed from October until March. However, also in summer months these processes are not infrequent. Cyclones bring to the Black Sea coast intensive precipitation with a high moisture content of wet tropical air, formed over the Mediterranean Sea. During wintertime, these cyclones advect not only abundant precipitation in the form of rain and snow, but also a «heat wave», causing intensive snowmelt and thus formation of floods of mixed-type genesis. This happened for example on March 13th, 2013 in the south of Sochi.

In the summer, cyclogenesis over the Mediterranean Sea becomes more active in conditions of large-scale northeasterly flow in the lower and central troposphere. Colder air moves over warm seawater areas and interacts with tropical air masses formed over North Africa. These conditions are very favourable for the formation of an explosive cyclogenesis over the Mediterranean Sea. Similar schemes of circulation are realized in various conditions of the large-scale atmospheric circulation.

First, there are abnormal northern locations of the Azores anticyclone. In recent years, the Azores anticyclone location is 1000–1500 km to the north of its normal climatic centre. In such a situation over the Mediterranean Sea, from the Iberian Peninsula to the Balkans, strong cyclogenesis (Fig. 2) becomes active. Small in size, but active cyclones move from the West to the East. A part enters the terrain of Turkey, but some move through the Black Sea to the Krasnodar and Stavropol territories, the Rostov region and the mid-Volga region.



Secondly, owing to the formation of a warm blocking anticyclone over Eastern Europe (over the water area of the Mediterranean Sea) northeasterly winds are formed. A distinctive feature of this synoptic process is "stopping" the Mediterranean cyclone over the Black Sea, because the inactive area of high atmospheric pressure blocks the cyclone path to the north or the northeast. Catastrophic inundations

were caused by such a cyclone on July 6-7, 2012 in Gelendzhik, Novorossiysk and the Krymsk district of Krasnodar territory (Fig. 2).

Thirdly, strengthening of zonal airflow and the formation of cyclonic waves over Mediterranean Sea can result in the abnormal southern position of the basic western jet 2. During the transit of cold atmospheric fronts of extensive Atlantic cyclones settling down over Eastern Europe or from accompanying quickly displaced deepening cyclones from the northwest, from Scandinavia,  abundant

precipitation falls out. Forced convection in a frontal region in combination with an orographic rise of warm and humid air forms powerful cumulonimbus rain clouds and, hence, strong downpours, thunderstorms and tornadoes. Such conditions led to inundations on August 22, 2012 in the Tuapse district.

In all above events, formation of so-called polar mesocyclones over the Black Sea waters was apparent – the axisymmetric vortex resembling tropical hurricanes. Since the first time this phenomenon was noticed in high latitudes, it is called «polar low». However, these cyclones are also often generated over the Mediterranean and the Black Sea. A polar low is a small, but fairly intense maritime cyclone that forms poleward of the main baroclinic zone (the polar front or other major baroclinic zone) (Rasmussen

E.A., Turner J., 2003). The horizontal scale of the polar low is approximately between 100 and 1000 km, that is, according to the Orlansky classification, a polar low is a phenomena of subsynoptic scale (the horizontal scale synoptic processes of more than 1000 km, and the same mesoscale processes of less than 100 km) (Markowsky and Richardson, 2002). The above definition can be extended, if necessary, by specifying the dominant physical mechanism responsible for the development of the low,

such as, for example a 'baroclinic polar low' or a 'convective polar low', the latter being driven primarily by organized convection (Rasmussen and Turner, 2003). A striking example is the weather



conditions prior to the flooding in Krymsk and Gelendzhik on July 6, 2012. The daily sum of precipitation was 150 mm in Krymsk (27 % of the annual norm), and 255 mm in Gelendzhik (36 % annual norm) (Arkhipkin et al., 2013).

In these synoptic situations, the probability of the formation of waterspouts is high. According to the meteorological station of Tuapse during 1946 - 1988, 46 cases of formation of waterspouts were recorded. From 2002 until 2011, the Federal Hydrometeorology and Environmental Monitoring Service recorded the formation of about 38 spouts. Waterspouts happen mainly in June - September over well-heated coastal regions, in warm and wet unstably stratified air, especially during the advection of cold and rather dry air by a cold atmospheric front. Spouts "descend" from cumulonimbus clouds. The lifetime of a waterspout is from several minutes to several tens of minutes, and they can pass a significant distance. In height, these spouts can reach several hundred meters, with diameters of order of tens of meters. There are eyewitness reports that waterspouts at Novorossiysk on August 8, 2002 had a diameter of 200 m and a height over 1 km. The spouts formed in the coastal region sometimes make landfall and move over mountain ridges. As a result, the seawater involved in the circulation of a spout, falls into river basins. It is impossible to resolve this process with standard observations; therefore, some experts are sceptical about a "fatal" role of waterspouts in the formation of powerful inundations (Sergin et al., 2001). Nevertheless, evidence of local residents and special research of the Krasnodar branch of Federal Hydrometeorology and Environmental Monitoring Service do not exclude such possibility. For example, after the August 8, 2002 event, the settlement Abrau-Dyurso suffered a catastrophic inundation when local residents found sea fish in their courtyards, and the river water had a saltish taste. The second example – strong downpours on June 20, 1988 (179 mm for 4 hours and 50 minutes) around Novorossiysk – was observed after the formation of a powerful waterspout over an open part of the sea and its "arrival" on the east coast of the Tsemessky Bay (Tkachenko, 2012). A part of this precipitation has probably been seawater. On the western coast of the bay, there was no rain during this day.



For the formation of high and dangerous floods, besides the amount of precipitation and their intensity (in the first few hours), the amount of precipitation for previous days and the degree of humidifying of the watershed is important. At strong downpours of 50–100 mm (and more) in 1–2 hours, the intensity of raising the water level in the rivers increases.

## 6  Features and regularities of flood routing

Formation of high and dangerous floods is probable in the upper and middle reaches of the rivers on the Black Sea coast. In a case of heavy rains in the lower river reaches and in a river mouth, flooding depends not so much on raising the water level in the river, but on powerful overland streams from the

slopes of the valley. Quite often at the lower reaches of a channel, flood formation starts earlier because the atmospheric water flux moves uphill from the sea, i.e. from the river mouth. On small rivers, a flood can therefore be observed almost simultaneously with powerful and destructive high floods.

Waves of floods in the mountains and in foothills move with great speed. It varies depending on the slope of the channels and the amount of water discharges. From the source to the mouth of the rivers

$Q_{max}$  increases, this allows a high speed of flood propagation, despite a reduction of the slope of the channels. The maximum flow velocities ($V_{max}$) during average height floods range from 1.5 − 2 m/s (Gostargayka and Dyurso rivers) to 3.5 − 4.5 m/s (the rivers to the south from city of Gelendzhik). The highest flow velocities vary from 5.5 to 7 m/s, but can even be higher. On October 7, 1970 a maximum velocity was measured on the Kuapse river of $V_{max}$ =8.75 m/s. Average flow velocities are almost 1.5

times less than the observed maximum. The velocity of propagation of flood waves is also less (Fig. 3), but because of the lack of reliable data this difficult to ascertain.

High floods at the Black Sea Coast rivers - due to the storm character of rainfall, large gradients of surface and rather small dimensions of the basins - are characterized by short duration, extremely fast rising and subsequent falling of the water level (Fig. 3). Floods, or their series, can last a number of

days. However, the main part of the flood wave passes, as a rule, within several hours − routinely no



longer than 0.5 – 1 days. However, the part of a flood that leads to flooding lasts even less. For example, the catastrophic flood on the Tuapse River in 1991 lasted ~4.5 days, its basic part, though, passed approximately within 1 day, and flooding of the floodplain lasted less than 4.5 hours (Panov et al., 2012). Residual flooding of floodplains remains longer.

The maximum rise of level ($\Delta H_{max}$) in the valleys of the Black Sea Coast rivers can reach 5 –7 m and even higher values (Fig. 4). The extreme water level rise is possible for catastrophic floods at parts of narrowing river valleys and channels, or upstream of bridges. The afflux component of rising water levels upstream of bridges and dams of wooden debris can be 0.5–2 m, but possibly exceeds this value. For example, during the catastrophic flood on August 1, 1991 between the settlements Kirpichny and

Tsypka at a reach of  the narrowing valley of the Tuapse river, the water level rise over pre-flood water levels amounted to 10 – 11 m, and upstream of the road bridge in the city of Tuapse ca. 10 m. At the hydrological gauging station of Tuapse a level of $\Delta H_{max} = 6.72$ m was observed. However, on the coastal rivers there are dominating parts where the rise of water level did not exceed a range from 1 to 3 m.

During abnormally high floods, practically the entire bottom of a river valley is inundated: the width of a stream very quickly increases in time, even by 15 – 20 times. For example, in 2012, during a flash flood, the Nechepsukho River in the settlement Novomikhailovskiy with normally a low flow channel of 20 – 50 m width increased at the time of flood to 300 – 700 m (Fig.5).

The flood wave undergoes its basic and final transformation at the lowermost reach of the river and in

the river mouth, i.e. downstream of the confluence of the last large tributaries. The conditions are an essential widening of the river valley, reduction of water slope, backwater effect from the sea (or from pebble and sand bars blocking the river mouth). As a result, the most dangerous flooding happens (Fig. 5 – 6) where the human settlements are usually found with basic resort and other infrastructure objects. The maximum depths of flooding by river waters reach more than 3 m, and by overland streams up to



0.5 m. After the inundation culmination, the largest part of water quickly flows down from flood plains into the river or directly into the sea.

The frequency of flooding of floodplains by river waters is controlled by the elevation of the area and protective dams, and the value and the probability of critical water levels and discharges. Vertical bed

deformations, artificial deepening of the channels, embankment and transformation of natural riverbeds in canals greatly influence the critical value of $Q_{max}$. Therefore, over time the values of $Q_{max}$ change. In 1970, near the village Gostagaevskaya, the water outflow into the floodplain occurred (in the absence of ice jams) already at $Q_{max}$ with a probability of 10 %, whereas now the probability (due to natural and artificial bottom improvement) is $Q_{max}$ <3%. In the early 1950s the floodplain near Tatyanovka village

(at the Psezuapse river) could be flooded at $Q_{max}$ with a probability <80%, in the early 1990s the probability was $Q_{max}$ <22 %. There are numerous such examples. This feature greatly complicates the development of effective methods of forecasting dangerous floods and resulting inundations.

In general, horizontal and vertical changes of river channels can reach significant values and create a definite threat for objects in the channel and on the banks. Thus, the flood, similar to the mudflow, on

the Matsesta River in September 1913, led to erosion of the riverbanks and shifted the shoreline by more than 400 m. During and after the inundation on the Tuapse River in August 1945, vertical channel erosion at the site of the gauging station and further downstream reached 2 m. The channel of the river Ashamba after a catastrophic high water on July 6-7, 2012 deepened by 2–2.5 m, and at the edge of the flood plain it became wider by 8 to 10 m (at places to 15m) (Evsyukov et al., 2013). Transit of a

catastrophic floodwater on the river Shirokaya Balka led on August 9, 2002 to an outwash of channel depositions 10 m deep, reaching the bedrock (Barinov, 2009). There are again many examples of this kind.

During floods together with the water, great volumes of deposits and debris are moved downstream. Quite often floods transform in mudflows possessing greater destructive ability, and result in other

hydro-morphological, economic and ecological adverse results. The main part of deposits accumulates



first on the flood plain before arriving at the lower reaches and at the mouths of the Black Sea coastal rivers. Therefore, besides destruction, the river water unloads a thick layer (10–20 cm and more) of deposits, of debris and refuse (Fig. 7a) on the flood plain. This is another aspect of the adverse effects of inundations together with channel deformations, deterioration of water quality in rivers and adjacent

seas. Deposits from floods and particularly mudflows seriously increase flood damage on developed terrain and of civil constructions.

Secondly, the significant part of deposits is accumulated in channels that lead to a reduction of their water transport capacity. If a channel is not periodically dredged, its transport capacity quickly diminishes. As a result, the frequency of dangerous flooding increases. It also has happened in the

settlement of Novomikhaylovskiy in 2010 and 2012, despite considerable protection measures against the inundations in the settlement in the 20th century, including high and continuous dams, and a wide and improved channel. Only after the last inundation, the channel cleaning has started (Fig. 7b). Thirdly, parts of the deposits remain in river mouths where these depositions quite often form a bar shoal which later is washed away during strong autumn-winter-spring storms. Other parts (very fine

sediment particle fractions) are carried away into the sea, forming a strongly pronounced turbid plume, unfavourable for recreational activities of resorts and deteriorating habitat conditions of aquatic organisms (Fig. 7c).

## 7    Temporal regularities of inundations

Despite a characteristic distribution of floods and of water flow within a year (table 1, Fig. 1), almost 71 % of all catastrophic and exceptional inundations took place in July - August (71 %) and in October - November (29 %). Some 52 % of large inundations happen in the summer and 26% in the September - October period. The principal reason of such distribution is the increase of extreme rainfalls in the warm season. According to meteorological records, nine rain events with a total precipitation > 100 mm/day

happened in the last 50 years in November-February while in May-October such downpours were



observed, at least, 46 times, and in 85 % of cases in June-September. In March-April, there are no records of such rainfalls. Besides, heavy rains in the cold season have a longer time duration than in the warm period. This reduces the probability of the formation of dangerous high waters. The contribution from water tornadoes formed in the coastal region from June until October can be one more factor for increasing flood levels.

On the contrary, the seasonal distribution of small and moderately dangerous inundations reflects, on average, a water regime of two groups of rivers of the coast – to the north of the Tuapse River, and to the south. It is characterized by sufficient uniformity. Some 30% of such inundations take place in winter, in the spring 12%, in the summer 28%, and in autumn 30%. The safest months with respect to inundations of all types are March (3.5 %) and especially April (1.5 %).

On longer time scales, we can observe a nonlinear and statistically insignificant trend of the increase of the number of inundations and, hence, of the expected damage as given in (Fig. 8a). It mainly is caused by a noticeable increase of the number of inundations in the period from the beginning of 1970s until the first years of 21th century. This positive trend can be challenged, but the objective reasons for it, nevertheless, exist.

First, these are the climatic changes observed in the region (Kononova, 2012; Panov et al., 2012; Sergin et al., 2001; Tkachenko, Volosuhin, 2013). The increase in water flow at a number of rivers (absolutely unequally distributed at the different rivers in this small territory), and mainly peak water discharges (especially in last quarter of 21th century) and maximum flow extremes (Fig.9) are considered as the hydrological reaction to these processes. This is true particularly, for example, for the increase of anomalously high peak discharges of water, such as in 1980, 1991, 1997, 2002, 2010 and 2012 and statistically significant (at $\alpha$ =5 %) violation of homogeneity of time series of the peak discharges (at the some rivers) in respect of the dispersion.

The climatic origin of the long-term dynamics of the number of inundations is obviously a combination of the number of inundations and the total (for a year) duration of southern longitude-type of





circulations of the Northern hemisphere (in B.L.Dzerdzeevsky's typification), characterized by the intensifying role of southern cyclones. According to N.K. Kononova (Kononova, 2012; http://atmospheric-circulation.ru/datas/), from the end of the 1950s on the increase of the duration of this type of circulation (Fig. 8b) is noted. In the early sixties, for the first time for 112 years of record

(from 1899 until 2012) southern longitude processes have exceeded their average number. An unprecedented growth of the duration of southern longitude processes has begun in the 1980s and only after 2000 started to drop, but they are still above the average level. That is characteristic, and the same dynamics can be found in the annual total numbers of precipitation in the region of the Krasnodar territory (Volosuhin and Tkachenko, 2013). Simultaneously, with weakening of the southern longitude

processes in the 2000s, a significant increase in the frequency of northern longitude processes is noted.

Secondly, the growth of the number of extreme inundations can be a consequence of wide scale and not always prudent economic activity. It often includes intensive construction works at flood plains and on alluvial cones of river mouths (Fig. 10), where 100 years ago there still was no activity, and in the late eighties, with the beginning of the 1990s where in most cases only temporary constructions and kitchen

gardens were established. The other reasons could be the termination (or decrease in scales) of works in the Post-Soviet period of dredging channels and maintaining protective dams in good condition (Fig. 7), and unreasonable and intensive land use on watersheds. A number of scientists connect the increase in the 20th century (in comparison with the 19th century) of inundations, mudflows, rock falls and landslides to the last factor. In general, the anthropogenic contribution to inundations for the Black Sea

coast is considerable, its effect constantly grows, breaking the relations between characteristics of inundations both of natural climatic and hydrological factors, and enhancing differences in their interannual variability, and finally increasing the magnitude of inundations. The last example is an inundation on July 7, 2012 in Krymsk. According to the prevailing weather conditions, it would have been considerable, but has outgrown any expectations due to a combination of several anthropogenic

factors. Evidence are the accumulation of the large volume of water in fish-breeding ponds and the headwater upstream of the bridge, the subsequent outbreak, the destruction of trees in the river basin;



damage was caused to unlicensed residential buildings in a region of potential flooding and the untimely information of the population. Other anthropogenic factors had an influence, too.

Without a thorough discussion of this point, it will be impossible to quantify and respond to hydrological hazards, and to predict changes of inundations in the future. Nevertheless, a number of scientists at present consider (Matveeva et al., 2013) that this tendency will continue. The data of the climate model ECHAM5/MPI-OM (scenario A2) highlight that during the summer season of 2046-2065 an intensive frontal region (one of the synoptic predictors of abundant precipitation) will be twice more often than in 1981-2000, and 3 times than in 1961-1980. For the winter season, we find a reverse relationship.

The second feature of the interannual dynamics of the number of inundations at the Black Sea coast of the Russian Federation is their recurrence with a duration of cycles from $6 - 7$ to $10 - 12$ years (fig 9a). The spectral analysis of the time series with the program STATISTICA10 (for five basic transformations and at different window width of the sliding average), has revealed the highest peak of the periodogram and spectral density for the period of ~8 years duration and essentially smaller in height at 7 years. Smaller peaks are found for the periods of 3.5, 5.0 and 23 and at 11–12 years. A similar recurrence was found in the number of inundations in the whole North Caucasus during 1980-2013 (Magritsky et al., 2013) and by V.A. Volosuhin and Ju. Ju. Tkachenko (Volosuhin, Tkachenko, 2013) in the change of the quantity of floods of category DP on the rivers of the Krasnodar territory. Therefore, a temporary reduction of the number and magnitude of inundations does not mean that in the near future (during a specified period) there will be no new complications of the situation with inundations. The tragic events of 2002, 2010 and 2012 show what can happen in cases of self-complacency.



## 8    Geographical features and hazard of inundations

Orographic features of the coast and features of rainfall, river-flow inundations and mixed type 1 inundations as a rule cover only a small number of local river basins and a limited area, especially in the case of rain showers of tornado-origin. Therefore, the spatial correlation of $Q_{max}$ for the coastal rivers is rather insignificant and quickly decreases with distance between watersheds. Within the first 50 km, the correlation coefficient ($r$) still can reach significant values – more than $0.6 - 0.7$ (at a wide range of fluctuations – from 0 to 0.9). Within $50 - 75$ km $r$ drops to $0.5 - 0.6$ and less, within $75 - 125$ km – $r \leq 0.5 - 0.4$, for distances of $125 - 150$ km and more $r$ decreases to $0.2 - 0.1$ and less. On average, the correlation for $Q_{max}$ ceases to be statistically significant (at the level of significance $\alpha = 5\%$ and the available duration of records) for the distances over $40 - 60$ km. The time of the maximum water discharges for a year on the Black Sea rivers and subsequent inundations sometimes coincides with $Q_{max}$. Inundations are observed on the rivers of the northern slope of the Caucasus, such as the rivers of Novorossiysk and Gelendzhik – with the rivers on the stretch from the Gechepsin River to the Afips River and on the rivers of the Tuapse district and the northern part of Sochi and on the rivers in the basins of the Psekups, Pshish and Belaya. The last observed event took place in July 2012. Catastrophic inundations can cover even larger extended areas, affecting even the southern extremity of the arid Anapa municipal district as in August 2002.

Cases of inundations vary at the Black Sea coast of the Krasnodar administrative territory irregularly, despite their rather small spatial dimensions. The least affected are Temryuk (not including the delta of the Kuban River) and the Anapa municipal districts, because there are mostly conditions of flat and foothill terrain, small amount of precipitation and rarely a channel network. Significant damage was recorded here from local storm rainfall floods: in Temryuk district, about six cases are known since 1972, in Anapa about 11 cases since 1960. In the summer of 2003 such an inundation caused destruction in the village of Taman with a damage of 2.5 million roubles. (Magritsky et al., 2003). Because of the low and flat coast between Anapa and the settlement Veselovka, the shallow sea near the



coast and the spoon-like shape of the shore, there is a certain potential danger of storm surges and of tsunami. In addition, numerous artificial water bodies are potentially dangerous according to (Panov et al., 2012). In the Anapa area they very numerous, nearly 39 in number and with the total area of 3.5 km$^2$.

In the Novorossiysk, Gelendzhik, Tuapse and Sochi municipal areas floods caused by extreme rain and powerful slope streams often lead to inundations. To a much lesser extent river-flow inundations result from snowmelt runoff (as in 1981, the rivers of Mezyb and Vulan; 2003 and 2013, the Mzymta river), or by breaking of dammed lakes (in 1968, the Mzymta river) and ponds (in 1977, the Mzymta river) or combination of several factors. Even more often, mass media report about local floodings of settlements

by storm waters and by slope streams. Additionally in this part of the coast, powerful floods are possible during wind-wave surges. Seaport infrastructure and objects of the resort-recreational economic sector, first of all, suffer from these. Similar events were noted, for example, in 1968 in the mouth of the Dagomys River, in 1992 on the reach between Sochi and Adler, in 2003 and 2009 in the mouth of the Mzymta River.

The average interannual repeatability of river-flow and of mixed type 1 inundations in the Novorossiysk, Gelendzhik, Tuapse areas and in Sochi is approximately one time in 2.1, 0.9, 0.7 and 0.45 years respectively. For comparison, in Anapa it happens once in 6 years, and for the whole Black Sea coast once in 0.3 year. The percentage of small and moderately dangerous inundations is equal to 87 % in Novorossiysk, Gelendzhik and Tuapse district. In Sochi, this percentage reaches 92 %, in

Anapa almost 100 %. Other cases are big, catastrophic and exceptional inundations. Most of all catastrophic and exceptional inundations, in total four, happened from 1945 until 2013 on the rivers of the Tuapse district.

In Sochi, large hazards of inundations are conditioned by the large area of the terrain, the number and length of the rivers, the higher amount of precipitation and the number of settlements. Most casualties

were recorded for the Novorossiysk, Gelendzhik and Tuapse districts. One of the factors of such regularity is the greater level of water level rise, inherent to the rivers of Novorossiysk, Gelendzhik and



Tuapse area (Fig.3); another is the fast formation and transit of floods on the rivers of these areas, because of their small dimensions, and often the mudflow character of the floods.

Most of all inundations happen in the lower reaches and the mouths of the Black Sea coastal rivers and accordingly greater damage and higher losses. Large economical activities and larger populations are concentrated there, and most of the factors of "spontaneous behaviour" of river, rainfall and seawater take place here. Therefore, a stricter approach is necessary for this terrain with a higher degree of scrutiny with respect to issuing permissions for the placement of social and industrial objects in this region (land use and land planning), for the estimation of cost of their insurance and of the protective action, and for the population evacuation in case of emergency.

During catastrophic inundations, the respective damage is high without depending necessarily on the dimensions of the terrain and the number of watersheds subjected to storm precipitation and rising water level in the rivers. The damage from the exceptional inundations in August 1991 was estimated at approximately 400 billion rbl. (or 680 mln. dollars, according to the official data of the Central Bank of the Russian Federation at exchange rates for different years), with 363 billion rbl. (615 million dollars) on the coast. The number of casualties reached about 40 people (including 11 people missing). The catastrophic inundation in August 2002 led to a damage of roughly 1.7 billion rbl. (54 million dollars) and casualties of ~60 (including missing people); in October 2010 there were damages of 2.5 billion rbl. (80 million dollars) and 24 deaths; in August 2012 some 1 billion rbl. (32 million dollars) and four people lost their lives. The exceptional inundation in July 2012 is not included in this list, as the main impact and the damage was in the Krasnodar territory, not at the Black Sea coast. During large inundations, human casualties also are possible; the size of direct material damage varies from several hundred thousand to several millions dollars, but, according to the available scarce data, has not exceeded 4 - 5 million dollars for the Black Sea coast rivers.

Between the magnitude of inundation and the value of direct material damage there is an obvious and explainable relationship where the value of the damage increases (when moving from small inundations to more destructive ones) as the curve is close to an exponential function, but with higher steepness



(Fig. 11). Certainly, the reliability of the relationship developed by the authors of the paper still is low; the confidence interval wide. The reasons are the small number of cases (17 values) and the low reliability of the initial data. However, similar relationships dictate essentially our understanding of the danger of those or other inundations.

In general, the economic annual risk of river-flow and mixed type 1 (river waters + rainstorm + slope streams) inundations can be estimated approximately for all the Black Sea coast of the Russian Federation at 13.2 million U.S. dollars, and the social risk at 2.1 human lives. In the area of possible flooding, the risk varies from 49 (data of the authors) to 74 settlements (open information of the Ministry of Civil Defence and Emergency Situations for the Krasnodar territory), and about 3100

residence houses and 18200 inhabitants.

## 9    Countermeasures for inundations and their efficiency

Economic and social losses from inundations at the Black Sea coast of the Krasnodar territory are one of the highest in the Russian Federation. Therefore, one important direction for safe and sustainable

development of this area is and remains the implementation of various actions for a reduction of this hazard. The last catastrophic events and the preceding conditions have highlighted weaknesses of traditional measures, consequences of their non-observance, and initiated the search of new solutions.

The basic conclusion from recent inundations consists of the need to consider (as objects of the efforts) not only the lower reaches and mouths of the Black Sea coast rivers where the main part of the social

and economic development of the coast is concentrated, but also total river basins and catchments (Magritsky et al., 2013; Sergin et al., 2001). This is the watershed, where its physiographic features determine the time of concentration of surface runoff water in river channels, the saturation of a river stream by suspended load, debris and other refuse, and the transformation of a common flood into a mudflow. Therefore, the countermeasures for inundations should necessarily include well-defined

actions on river watersheds. These measures should include steps directed to the improvement of the



water controlling ability of reservoirs (by means of stopping woodcutting in mountain forests, optimization of composition of forest vegetation et al.) to the reduction of erosion of bedrocks, to reduction of littering of slopes of river valleys and riverbanks.

Definitely, basic measures are necessary in the middle and especially in the lower reaches of the rivers.
Among engineering measures, for a long time and successfully carried out here, these are the construction of bank dams, deepening and improvement of river channels, and reinforcement and protection of riverbanks against flood erosion. For example, out of 48 rivers discharging to the sea between Tuapse and the settlement of Leselidze 20 channels, i.e. 42 % were improved. But these measures are only effective, if the embankments/dams are in good condition and of sufficient height, the
river channel is constantly cleaned from depositions and debris, the channel improved and maintaining sufficient transport capacity, not only for water but also for the considerable quantity of deposits, debris, and that this capacity remains maintained under the bridges. Default, or infringement of these and other requirements lead to those consequences, which all could observe, for example in the settlement Novomikhaylovskiy during the large inundations in 2010 and 2012.

At the same time, dredging channels, the removal of boulders, pebbles, gravel and sand from a channel and from floodplains should be carried out taking into account possible adverse consequences of this action. Accompanying this action, there are many that can overturn any positive effect: - decrease of the low flow water level in rivers, - lowering of ground water level, - the almost inevitable water supply infringement, - undermining and infringement of the stability of hydraulic constructions in the channel
and on the banks, - change of the balance of beach-forming deposits and intensifying of the vulnerability of sea beaches.

Other engineering measures such as the increase in height of dams around the objects of importance, channel "replacement" as in a case in the lower reach of the Tuapse River, maintenance of free drainage or filtration of rainwater in inhabited terrains are also suggested. At present, the operative practice of
"artificial dambreaking" of coastal barriers in the river mouth is rarely applied. Coastal barriers formed by sea waves and storm surges block the river channel and do not allow the river waters to flow freely



into the sea. At the approach of the flood wave, the coastal dam, which routinely protects the mouths of the Black Sea rivers from the wind induced sea surges, is at the initial moment of the development of inundations a serious obstacle for the free discharge of river water into the sea, i.e. being one more factor to contribute to inundation.

Regulating the maximum flow by water reservoirs on the Black Sea Rivers is ineffective owing to the impossibility of building in this region large regulating storage capacity.  They fill up fast with sediment, including landslip and mudflow deposits. Additionally there is a high danger of destruction of the dams because of the high seismicity of the terrain. Where artificial reservoirs nevertheless are built, they are an additional factor leading to powerful nature-anthropogenic river-flow inundations.

Therefore, such hydraulic engineering structures should be constructed under multi-hazard aspects and designed very reliably.

Among non-engineering approaches, it is necessary to pay attention, first, to an increase of efficacy of the preliminary forecasts of the maximum water levels and discharge, the timely warning of the population and subjects of economy about the approach of "the big water". Modernization of the hydro-

meteorological monitoring system for this purpose is required. The first steps in this direction have already been made by the Ministry of Emergency and Civil Defence in the Krasnodar territory. Since November 2012, the computerized system of monitoring flood situations on the rivers and reservoirs (http://test.emercit.com/overall.html) in the region is in place. The installation of several rainradar-tracking stations for monitoring the intensity and quantity of precipitation is required. Vulnerability

assessment and increasing of preparedness of local population are also important aspects among measures of flood risk minimization (Zemtsov, 2014).

Secondly, a clear understanding of the reasons, features and systematics of the origin and development of inundations and their adverse hydrologic-ecological, morphological, social, and economic consequences is necessary. Here are good prospects for numerical modelling and GIS-technologies

(Fig. 12). Modelling of water and debris flow is required to estimate flooding borders, water levels,



depths and flow velocities in key areas. Such data could be the base for a detailed hazard assessment and zonation of the river valleys (Petrakov et. al., 2012).

Thirdly, restrictions (by various means – from administrative measures to flexible flood insurance) are necessary for the processes of developing the territory to reduce its potential flooding hazard. For this purpose, these limits (for inundations different in their dimensions) should be made known and the terrains differentiated at the degree of their hydrological hazards. This will be required for solid land-use planning and planning permissions.

## 10  Conclusions

The list of the dangerous natural phenomena at the Black Sea coast of the Krasnodar territory of the Russian Federation is extensive, but inundations cause the greatest damage. Such situation arises from the influence and interactions of many different factors. Among natural factors are the specific location of the area, the complex orography of the territory, the high drainage density, small basins area, and the big water slopes and weak regulating ability of river watersheds. An important role is played by the large quantity and the extremeness of rainfall, and the intense flood regimes of the rivers. Extreme floods form rapidly and transit fast downhill. That leads to a fast and substantial increase of water levels, the frequent transformation of rain floods into mudflow-like streams, and the contribution of powerful storm and overland streams to additional terrain flooding. Among anthropogenic factors are the location of the main part of settlements, objects of the industry, social sphere and the resort industry, the transport infrastructure in river valleys and the mouths of the Black Sea coastal rivers.

By genesis at the Black Sea coast, inundations are generated by river-flow and river-flow–rainstorms (mixed type 1). They dominate in number, repeatability and damage values. At the coast and in river mouths the inundation can be caused also by storm surges, or by storm wave induced surges, and the interaction of river and the sea. We can distinguish inundations by terrain coverage (the number of involved watersheds and rivers) and intensity, and by magnitude of the damage - small, moderately



dangerous, large, catastrophic and exceptional inundations. The probability of their occurrence accordingly is ~20, ~10, ~4–5, ~2–2.5 and <1 % respectively.

The floods, which lead to river-flow and mixed type 1 inundations, are formed by abundant and heavy storm rainfalls – at the transit of southern cyclones, cold atmospheric fronts of extensive Atlantic cyclones covering Eastern Europe, or accompany cyclones quickly arriving from the northwest. As a special case, falling into categories of "cloud burst», rainfall discharges as a result of the destruction of sea-born tornadoes on land. The mountain relief has a considerable role in the formation of abundant precipitation. Catastrophic inundations are generated by an abnormal combination of synoptic processes and convective phenomena.

The characteristic features of dangerous floods are their rapid formation and propagation, a significant increase of water level (up to 5–7 m and more) and multiple increase of water discharges (at times practically from values close to zero to several hundred m3/s and even >1000 $m^3$/s). During floods, practically the entire bottom of the river valley can be submerged, therefore all this terrain becomes a region of significant risks for land use and management. The flood wave undergoes main and dangerous transformations at the lowermost reach of the channel and in the river mouth where settlements and basic economic objects routinely are developed. The maximum heights of flooding by river waters can reach here 2 – 3 m, and in slope streams 0.5 m. Flooding last routinely some hours; residual floods continue longer. Floods carry together with the water great volumes of sediments and debris. Quite often floods transform into mudflow streams with a high destructive capability, and have, different from floods, severe hydrological-morphological, economic and ecological consequences.

Most catastrophic and exceptional inundations arise in the summer and in the early autumn. Small inundations during the remaining year reflect the seasonal distribution of river flow and floods in the Black Sea Rivers.

On the interannual scale, the increase of the number of inundations and, hence, the damage involved is implicit. It mainly is caused by an appreciable increase of the number of inundations in the period from



the early 1970s until the early years of the 21th century. The main reason is found in a changing climate but another influence, especially regarding the extreme inundations, are anthropogenic such as irrational and badly planned economic activities in channels, flooded terrains, and on river watersheds. The mean annual frequency of inundations and dangerous floods on the entire coast is about once in 0.3 years. The

number of inundations in the region varies with a duration of cycles from $6-7$ to $10-12$ years.

Not all administrative areas of the coast are equally in danger and vulnerable to river inundations. The most dangerous areas are the Novorossiysk, Gelendzhik, Tuapse and Sochi municipal districts. The larger Sochi area is in danger of a high frequency of inundations, whereas in the other three areas the high danger results from higher extremes of storm rain floods. Not surprisingly, here are more cases of

catastrophic inundations and loss of human lives. In general, the total annual economic and social risk from river inundation can be approximately estimated at 13.3 mln. U.S. Dollars and two human lives at the Black Sea cost of the Russian Federation.

Our systematic analysis will increase awareness of the public to raise the level of safety and security of the terrain, and objects and population in the Black Sea Coast area not only by improving engineering

actions, but also by the optimization of the terrain, and the increase of system effectiveness of the monitoring and forecasting critical hydrometeorological situations, resulting in improved early danger warnings.

## Acknowledgements

The authors are grateful for the support of their data base collections, field investigations, and hazard estimations as members of the Natural Risk Assessment Laboratory (NRAL) of Lomonosov Moscow State University under the grant no. 11.G34.31.0007. Analysis of floods characteristics, features and regularities of flood routing, and mapping of flooding zones were financially supported by the Russian Science Foundation (grant No. 14-17-00155). Part of the presented research related to synoptic

conditions analysis was supported by grant of RFBR 13-05-41058. Inundation typification,



methodology of flood cases identification based on hydro-meteorological data, and the design of countermeasures were developed as part of the grant of the Russian Science Foundation (grant No. 14-37-00038).





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



Table 1. Average water discharges and season distribution of water flow of the rivers of the Black Sea coast of Krasnodar territory

| The river, gauging station | The basin area, km$^2$ | Average altitude of basin, m | The period of observation | Average water discharge (m$^3$/s)$_{1)}$/variation coefficient$_{2)}$ | Season distribution of water flow (%) | | | |
|---|---|---|---|---|---|---|---|---|
| | | | | | spring | summer | autumn | winter |
| Gostagayka, Gostagaevskaya | 106 | 160 | 1949–2000, 2009–2011 | 0.36/0.59 | 39.8 | 10.8 | 6.2 | 43.2 |
| Dyurso, Abrau-Dyurso | 51.9 | 190 | 1948–1976 | 0.47/0.36 | 32.3 | 7.2 | 6.3 | 54.2 |
| Mezyb, Vozrozhdenie | 100 | – | 1983–1994 | 2.0/– | – | – | – | – |
| Aderba,Sveltyi | 57.4 | 250 | 1966–1993, 1996–1997 | 0.94/0.36 | 32.5 | 7.6 | 7.7 | 52.2 |
| Vulan, Arkhipo-Osipovka | 265 | 240 | 1948–2006, 2009–2011 | 5.71/0.36 | 28.5 | 8.3 | 9.5 | 53.7 |
| Tuapse, Tuapse | 351 | 390 | 1938–1942, 1944,1945, 1949–1996,2009–2010 | 13.1/0.30 | 28.3 | 7.0 | 18.0 | 46.7 |
| Ashe, Ashe | 282 | 570 | 1956–1982, 1984–1991 | 14.3/0.29 | 30.5 | 9.3 | 18.9 | 41.3 |
| Kuapse, Mamedov Shchel | 14.6 | 380 | 1946–1991, 1993,1994, 1996–2011 | 0.74/0.35 | 31.5 | 9.9 | 16.78 | 41.9 |
| Psezuapse, Tatyanovka | 255 | 760 | 1955–1992 | 14.4/0.20 | 35.0 | 11.2 | 17.9 | 35.9 |
| Shakhe, Soloh-aul | 423 | 1010 | 1926–1991, 1993–1994, 1998–2001,2003–2011 | 28.4/0.22 | 39.7 | 16.3 | 18.0 | 26.0 |
| Psiy, Tuh-aul | 20.4 | 700 | 1946–1988 | 1.20/0.20 | 34.6 | 12.4 | 17.6 | 35.4 |
| West Dagomys, Dagomys | 49.0 | – | 1974–1994, 1996–2003, 2004–2011 | 2.43/0.27 | 29.5 | 10.2 | 22.0 | 38.3 |
| Sochi, Plastunka | 238 | 840 | 1927–2011 | 15.2/0.22 | 38.2 | 15.6 | 18.8 | 27.4 |
| Sochi, Sochi | 296 | 720 | 1945–2012 | 16.5/0.21 | 37.1 | 14.4 | 18.8 | 29.7 |
| Khosta, Khosta | 98.5 | 480 | 1927–1994, 1997–2011 | 5.06/0.20 | 31.2 | 14.4 | 19.8 | 34.6 |
| Mzymta, | 510 | 1670 | 1945–1994,1996– | 34.5/0.16 | 36.9 | 33.7 | 16.6 | 12.8 |



| Krasnaya Polyana Mzymta, Kazachiy Brod | | | 2002, 2010–2012 | | | | | |
|---|---|---|---|---|---|---|---|---|
| | 839 | 1340 | 1967–2012 | 54.5/0.18 | 36.4 | 26.4 | 18.7 | 18.5 |

The note. 1) Data at some stations extended to longer period of observations.

2) corrected for a negative shift




Table 2. Maximum water discharges of the rivers of the Black Sea coast

| The river, gauging station | The period of observation | Peak water discharge ($Q_{max}$), m³/s | | | Water discharges, m³/s, and their probability[2] | | | |
|---|---|---|---|---|---|---|---|---|
| | | medium $Q_{max}$ | coefficient of variation[1] | max. $Q_{max}$ /year | 1% | 2% | 5% | 10% |
| Gostagayka, Gostagaevskaya | 1949–2000, 2009–2011 | 12.9 | 0.84 | 43.8/1966 | 53 | 44 | 34 | 27 |
| Dyurso, Abrau-Dyurso | 1948–1976 | 16.5 | 0.65 | 38.3/1967 | 49 | 44 | 37 | 31 |
| Aderba, Sveltyi | 1966–1993,1996–1997 | 63.1 | 0.82 | 178/1981 | 240 | 207 | 165 | 130 |
| Vulan, Arkhipo-Osipovka | 1948–2006, 2009–2011 | 367 | 0.50 | 1050/1980 | 1030 | 870 | 700 | 580 |
| Tuapse, Tuapse | 1938–1942,1944,1945, 1949–1996,2009–2010 | 450 | 0.90 | 2300/1991 | 2330 | 1790 | 1100 | 660 |
| Ashe, Ashe | 1956–1982, 1984–1991 | 325 | 0.72 | 1435/1991 | 1340 | 1070 | 750 | 535 |
| Kuapse, Mamedov Shchel | 1946–1991, 1993, 1994, 1996–2011 | 34.1 | 0.62 | 115/1991 | 125 | 105 | 81 | 63 |
| Psezuapse, Tatyanovka | 1955–1992 | 320 | 0.59 | 1200/1991 | 1010 | 830 | 645 | 525 |
| Shakhe, Soloh-aul | 1926–1991, 1993–1994, 1998–2001,2003–2011 | 275 | 0.50 | 938/1982 | 905 | 750 | 555 | 435 |
| Psiy, Tuh-aul | 1946–1988 | 28.3 | 0.62 | 88.7/1956 | 93 | 80 | 62 | 49 |
| West Dagomys, Dagomys | 1974–1994, 1996–2003, 2004–2011 | 140 | 0.79 | 511/1997 | 565 | 480 | 355 | 265 |
| Sochi, Plastunka | 1927–2011 | 282 | 0.45 | 719/1997 | 715 | 625 | 515 | 440 |
| Sochi, Sochi | 1945–2012 | 355 | 0.50 | 990/1997 | 975 | 860 | 710 | 590 |
| Khosta, Khosta | 1927–1994,1997–2011 | 175 | 0.51 | 458/2002 | 485 | 420 | 345 | 287 |
| Mzymta, Krasnaya Polyana | 1945–1994,1996–2002, 2010–2012 | 176 | 0.34 | 360/1997 | 365 | 330 | 287 | 253 |
| Mzymta, Kazachiy Brod | 1967–2012 | 360 | 0.38 | 730/2003 | 800 | 720 | 615 | 535 |

The note. [1] corrected for a negative shift, [2] based on probabilities distribution by Kritsky-Menkel and Pearson of type III. The crossed out section indicates lack of data.




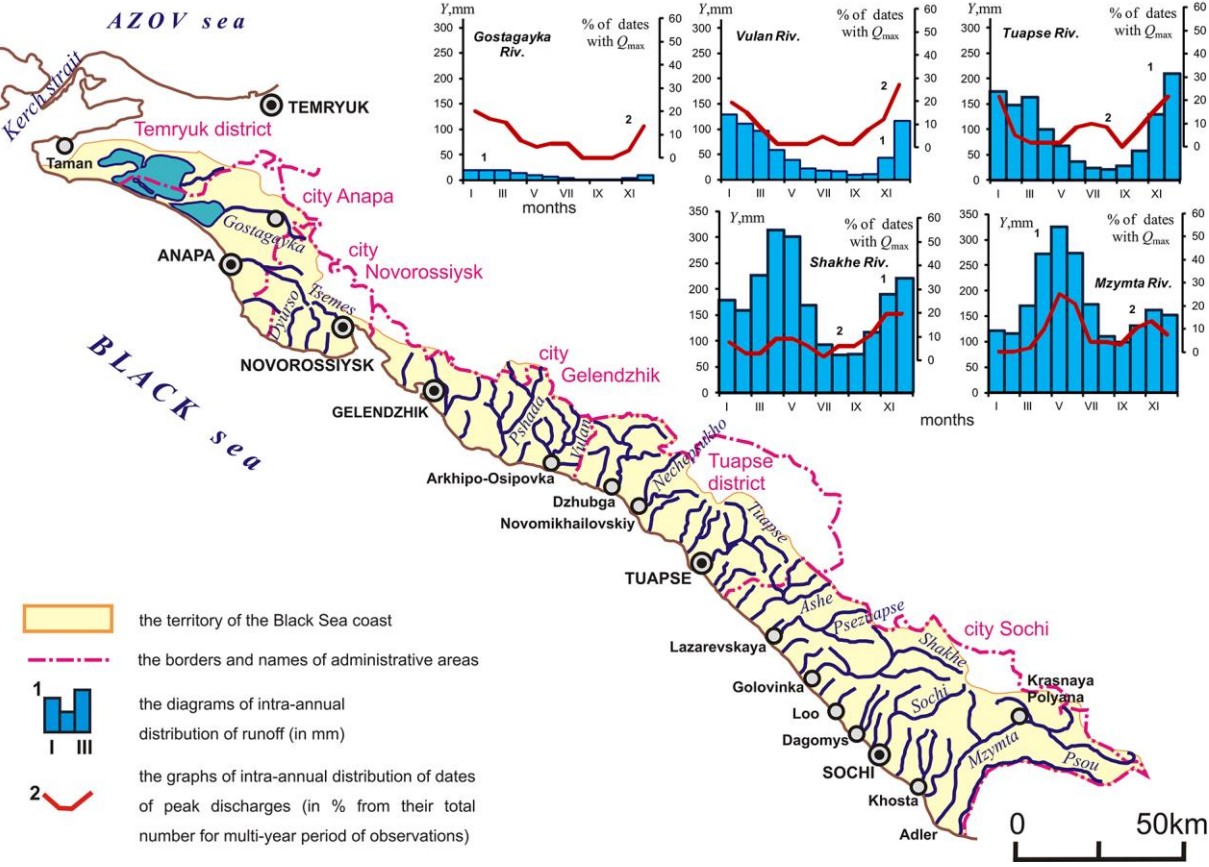

Figure 1. Hydrography of the Black Sea coast of Krasnodar territory of the Russian Federation and the water regime of the rivers.



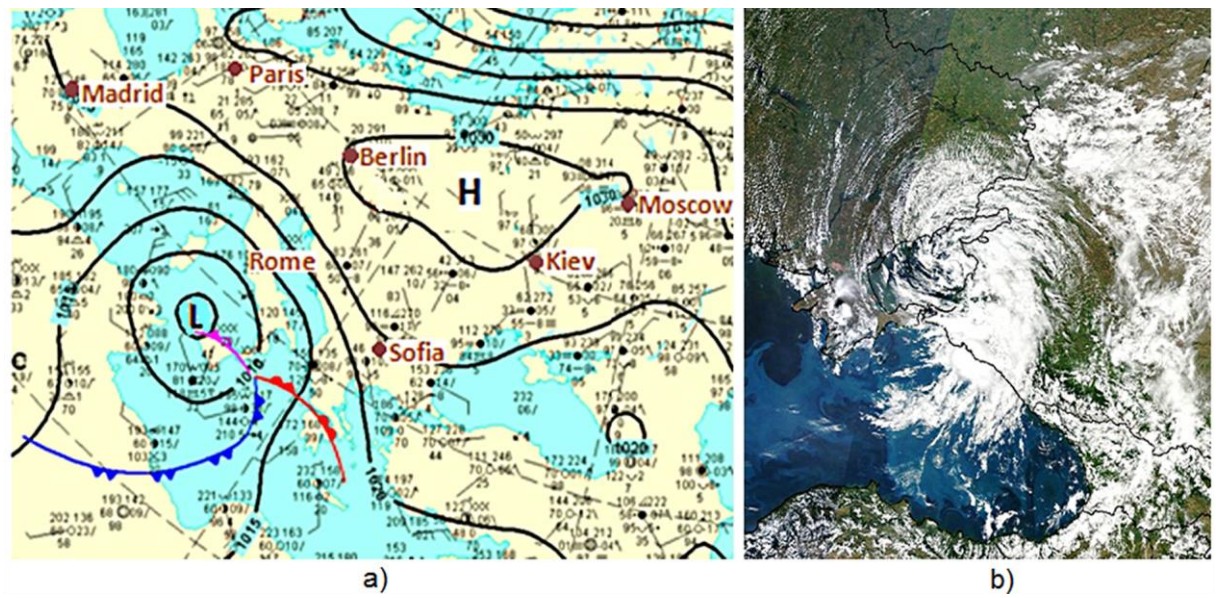

Figure 2. Different examples of synoptic processes: a) typical Mediterranean cyclone (marked with the symbol «L») and an Atlantic anticyclone (marked with the symbol «H») on a bottom pressure analysis map; b) the mesocyclone which has invoked catastrophic inundations in Krymsk in July, 2012

5    (NOAA/Goddard Space Center – NASA EOSDIS LANCE-MODIS (http://lance-modis.eosdis.nasa.gov/imagery/)).



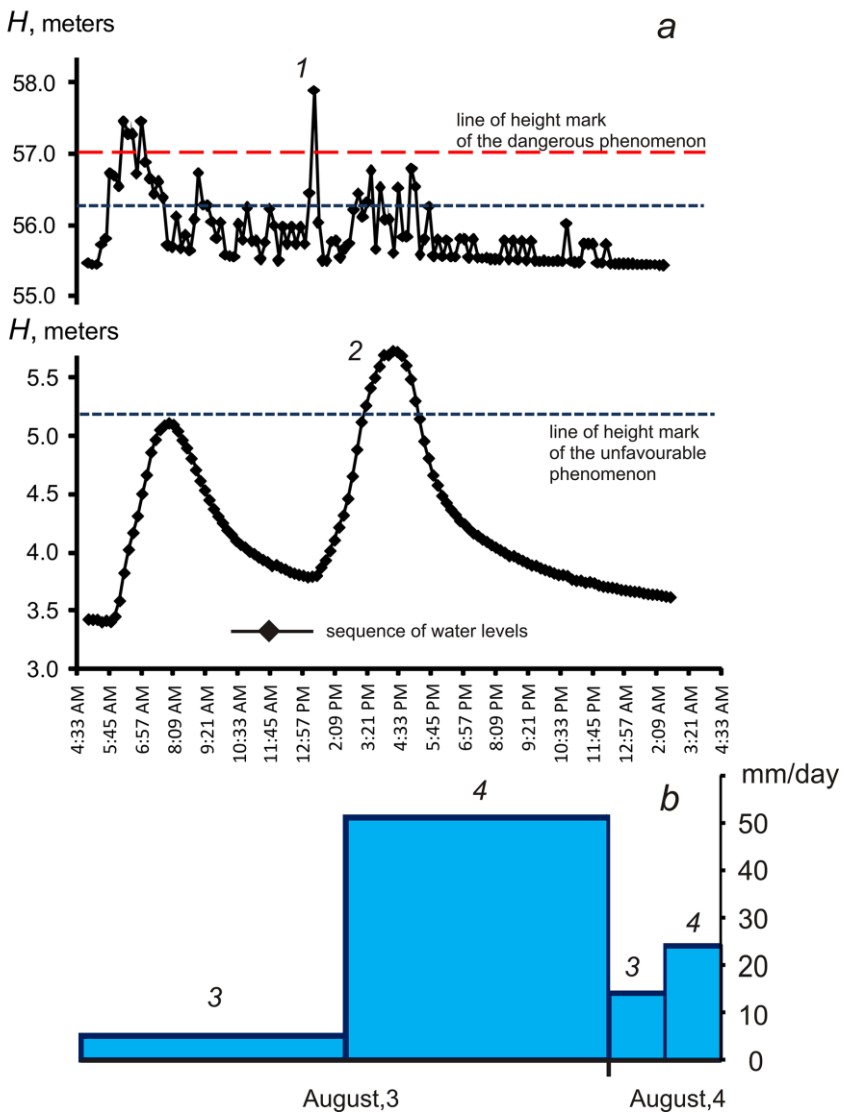

Figure 3. Graphs of fluctuations of water level of the Vulan River (a) and the diagram of the amount of daily rainfall (b) from August 3 - 4, 2013. Symbols: 1 – gauging station of Tekos (at 13 km upstream of the river mouth), 2 – gauging station of Arkhipo-Osipovka (in a river mouth); 3 – rainfall at meteorological station Gelendzhik, 4 – rainfall at meteorological station Dzhubga. Levels are relative to gauge-datum. The velocity of the flood was 2.4 km/s (at a longitudinal gradient on the reach of 0.004)





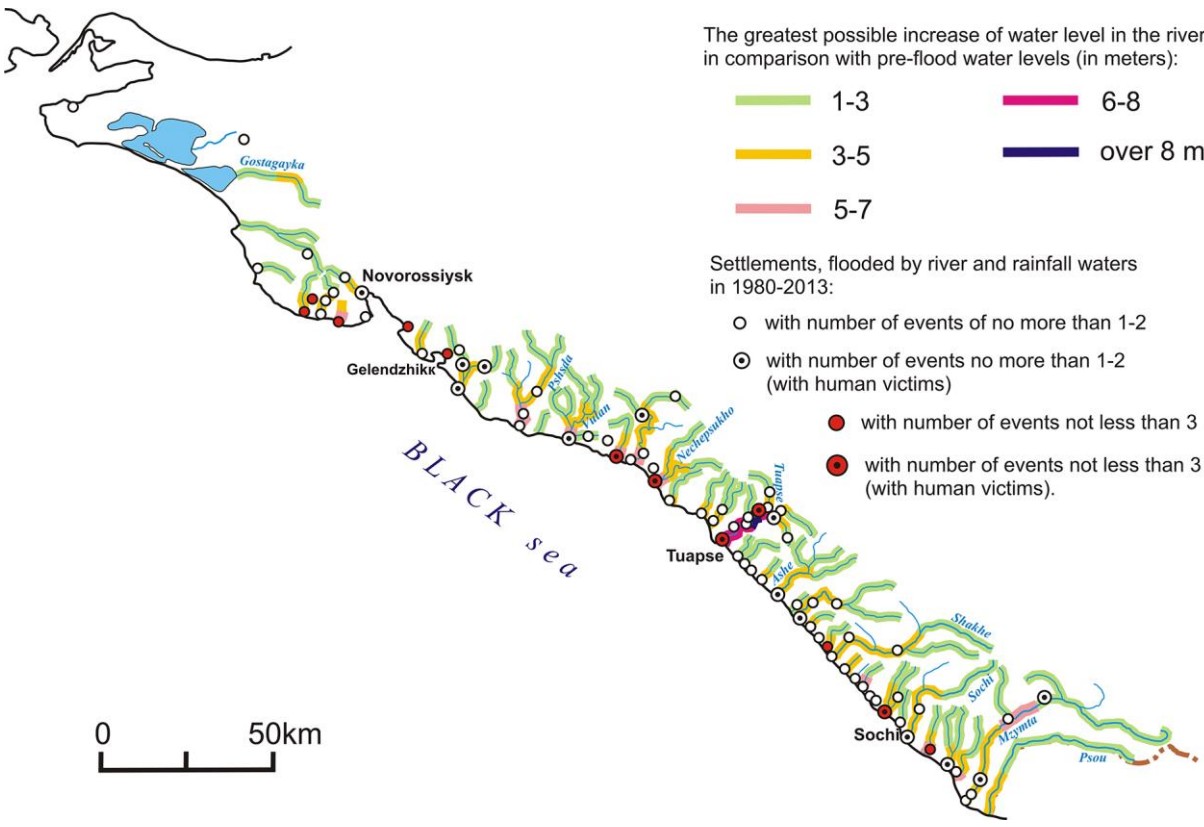

Figure 4. The Black Sea coast of the Russian Federation: difference in value of the greatest possible increase of water levels in river channels, and after-effects of inundations.



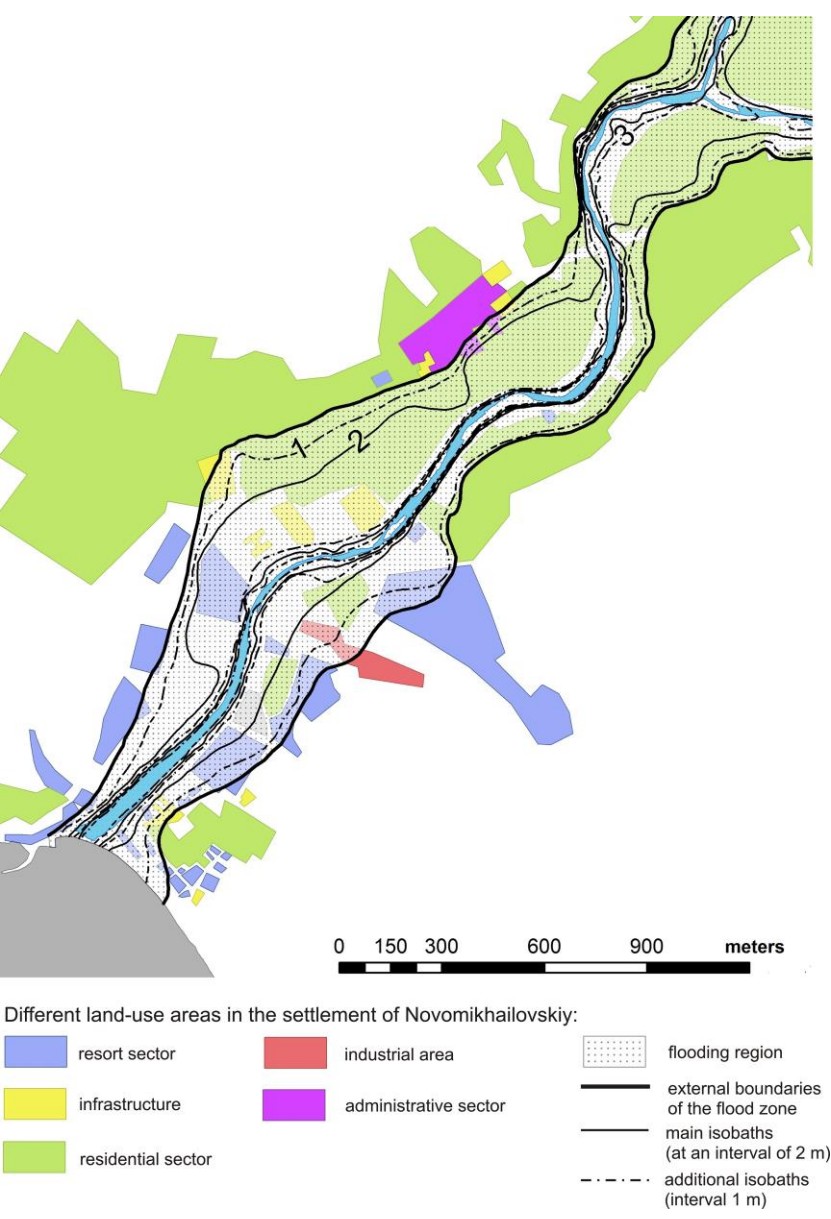

Figure 5. Reconstruction of borders and depths during the river-flow inundations in the settlement of Novomikhailovskiy in 2010 and 2012 (Magritsky et al., 2013).


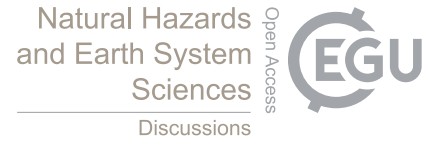

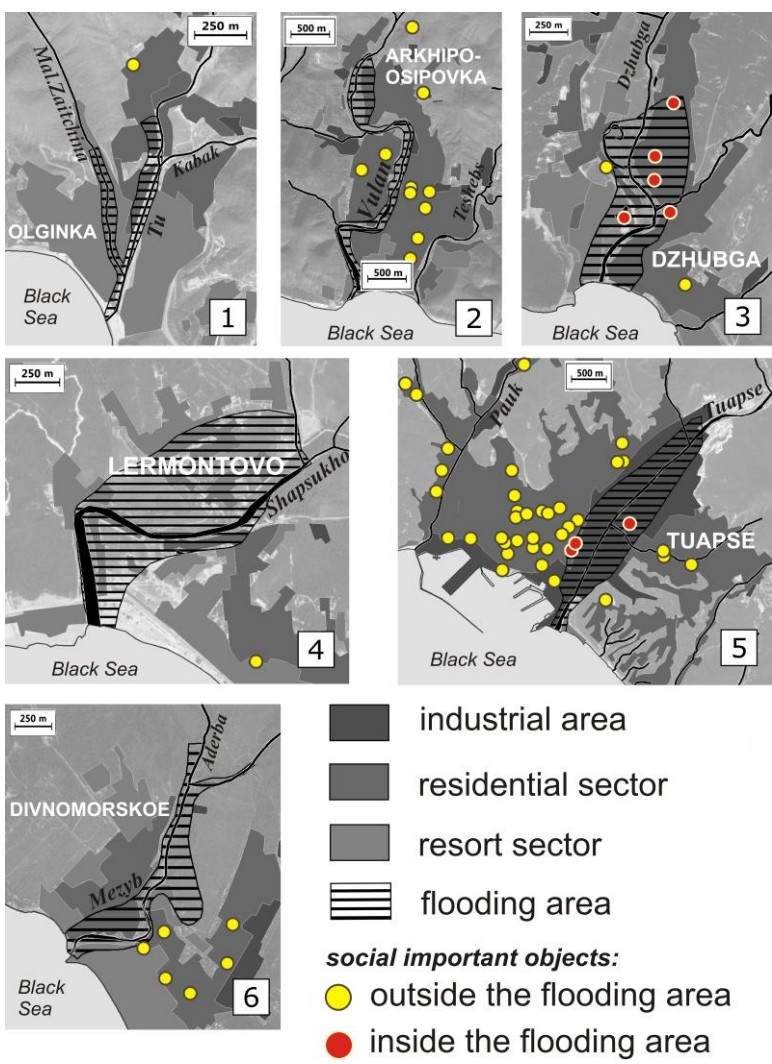

Fig. 6. Borders of flooding zones in the mouths of the rivers Dzhubga (settlement Dzhubga; area of flooding $F \approx 0.79$ km$^2$), Mezyb (settlement Divnomorskiy; $F \approx 0.44$ km$^2$), Shapsukho (settlement Lermontovo; $F \approx 0.83$ km$^2$), Nechepsukho (settlement Novomikhailovskiy), Tuapse (Tuapse; $F \approx 2.08$ km$^2$), Vulan (settlement Arkhipo-Osipovka; $F \approx 0.73$ km$^2$), and Tu (settlement Olginka) during the big and catastrophic inundations.



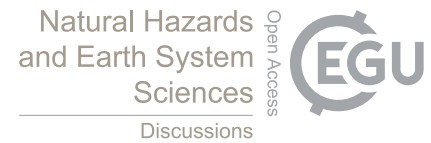

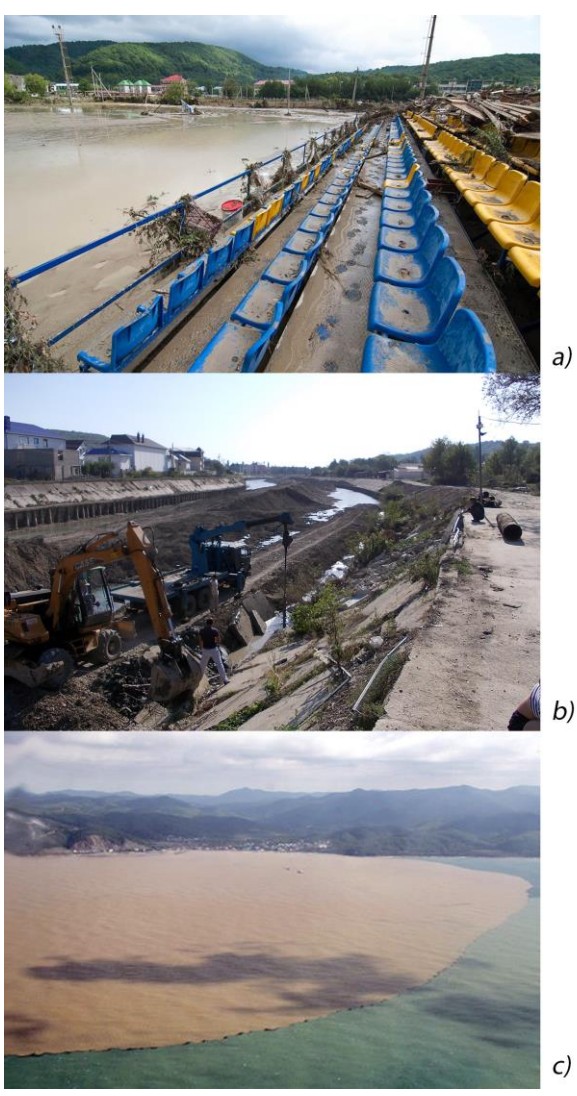

Figure 7. Consequences of a catastrophic high water in August 2012. Settlement Novomikhaylovskiy (Tuapse district). a -depositions of river sediments and refuse at in the city stadium (August, 2012; www.livekuban.ru); b -clearing the channel from deposits and vegetation (October, 2012, D.V.Magritsky); c - «turbid plume» in the mouth of the Nechepsukho river (August, 2012; http://ria.ru/)



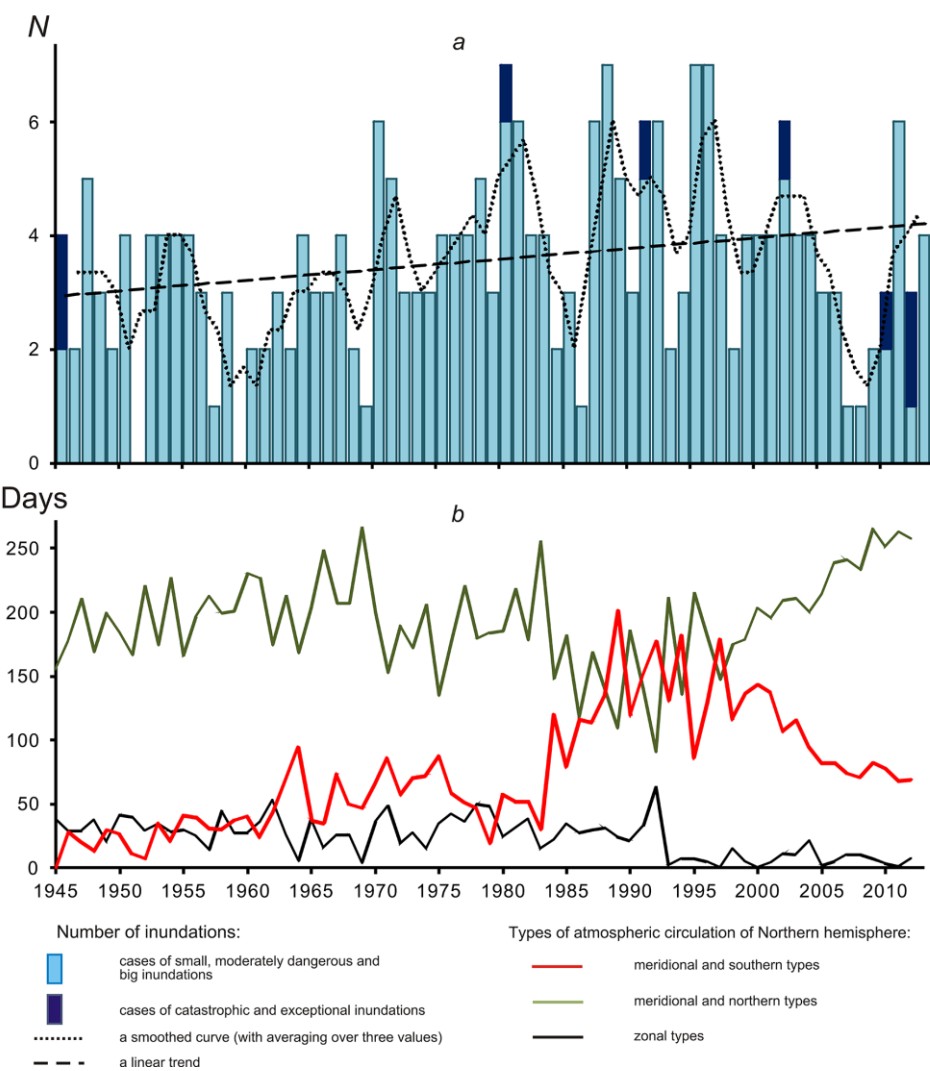

Figure 8. Interannual changes of the number of inundations at the Black Sea coast of Krasnodar territory (a) and fluctuation of total duration for a year of groups of types of atmospheric circulation of Northern hemisphere in B.L.Dzerdzeevsky's typification in 1945-2013 (b; by data from site: http://atmospheric-circulation.ru/datas/).



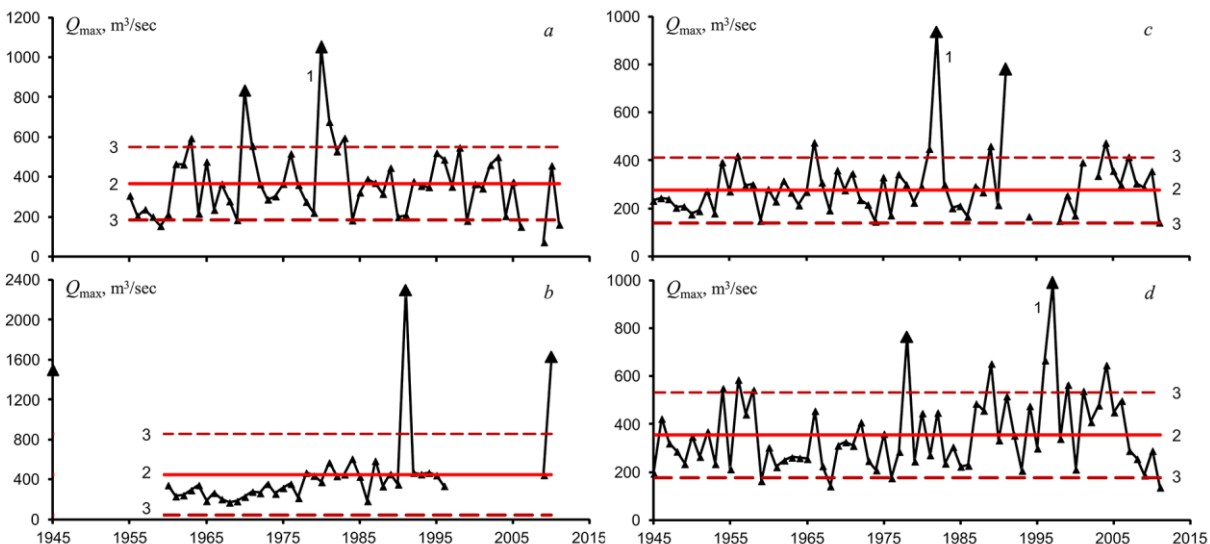

Figure 9. Long-term fluctuations of the maximum water discharges of the rivers Vulan (a), Tuapse (b), Shakhe (c) and Sochi (d): 1 – maximum water discharges, 2 –mean annual water discharge, 3 – maximum deviation from the mean annual water discharge by 1 σ.




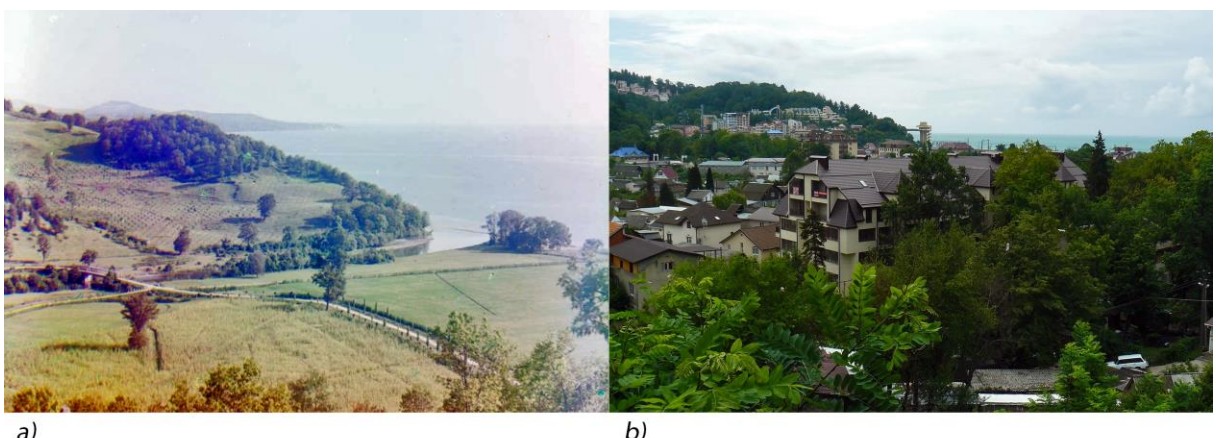

Figure 10. Mouth of the Dagomys River (the city-resort of Sochi) in the beginning of the 20th century and in the beginning of the 21st century. The left picture (a) – S.M. Prokudin-Gorskiy's photo (1910-1915, Library of the Congress of the USA). The right picture (b) – a photo from a resource http://www.panoramio.com –user №6172839l (on August 18th, 2013)

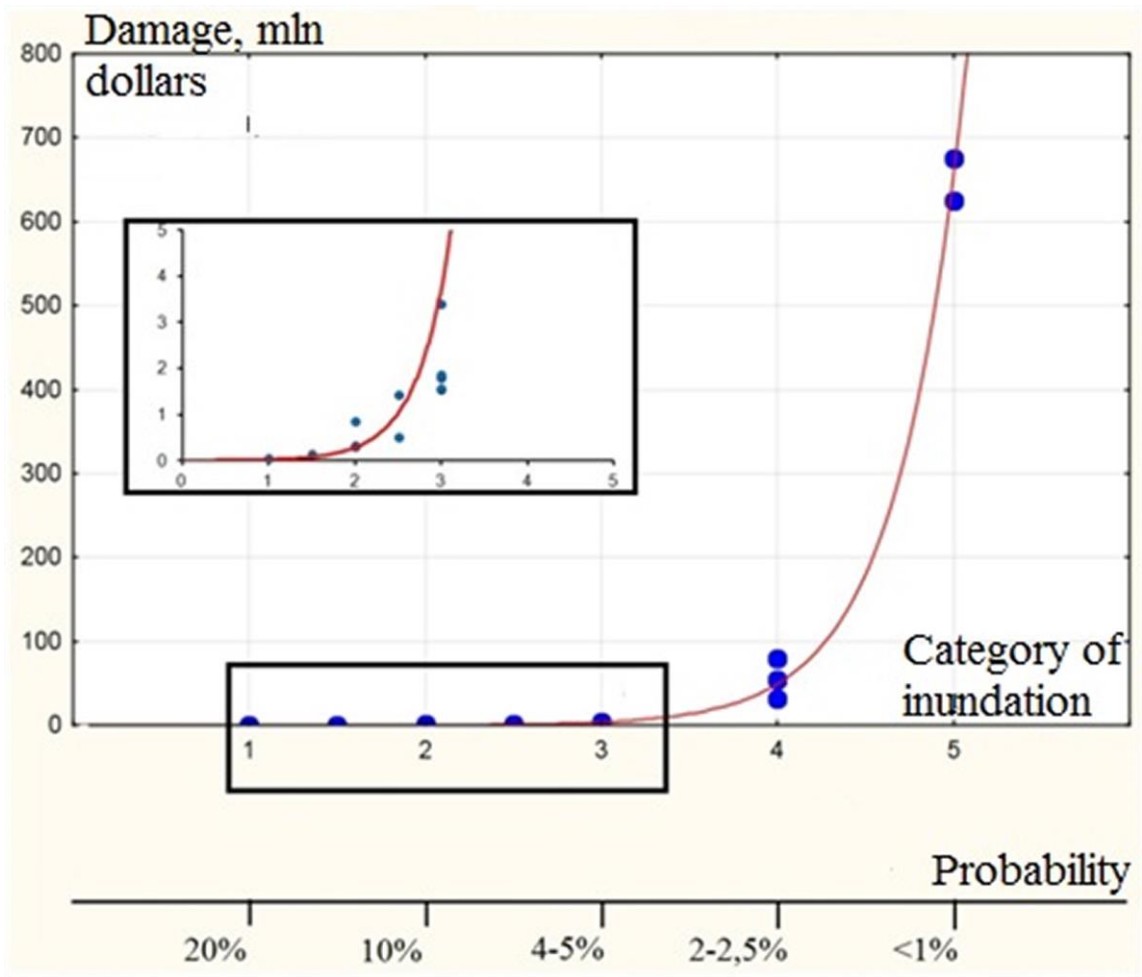

Figure 11. Empirical relationship between direct material damage (millions of US dollars) and type of inundation (river-flow and mixed type 1), or probability of $Q_{max}$





Figure 12. An example of numerical modelling of inundation in the valley of the Ashamba River (terrain of city-resort Gelendzhik): a) terrain before flood, b) calculated maximum flood depths