# Peer review of "Causes and systematics of inundations of the Krasnodar territory on the Russian Black Sea coast"

_Natural Hazards and Earth System Sciences, 2015_

## Referee Comment (RC1) · S. Fuchs (Referee) · 4 Feb 2016

S. Fuchs (Referee)

sven.fuchs@boku.ac.at

The authors present an overview on inundation hazard and risk for the Russian Black Sea coast, spanning from flood hazard assessment and the roots causes of inundation to elements at risk and losses. Comprehensive overviews on natural hazards and risk in Russia are generally underrepresented in the English scientific literature so far. As such, this is a topic of considerable relevance to the readers of Natural Hazards and Earth System Sciences.

Therefore, the work should be considered for publication. However, some of the content needs additional efforts before this manuscript may become acceptable for publication. Please find my remarks below.

Abstract, line 13: insert a "the" before Krasnodar
[Figure]

Abstract, line 16 ff: the verb is missing in this sentence

Introduction, first and second para: please provide evidence for the numbers given, e.g. references

Objectives. . .: please check the use of "km2" (same page 6, lines2 and 3)

Objectives. . ., line 14: please provide evidence for the number given, e.g. a reference

Hydrological data. . ., line 22: replace "for days" by "diurnal". This paragraph describes data which is not represented in Table 1. Could you please add this information to Table 1?

Inundations. . ., line 16: what is meant by the term "mastered terrains" (same page 11, line 5)?

Inundations. . ., line 19/20: should be "Alekseevsky and Magritsky (2013)"

Inundations. . ., page 10, lines 1 and 23: please explain the type 1 inundations a bit more, and take care to use the same wording ("Inundations of mixed type (No1). . ." versus "mixed type 1 inundations" since it makes the text more accessible when using similar expressions.

Page 13, line 8: please provide a short explanation on the "western jet 2".

Page 13, lines 19/20: Please check reference.

Page 15, lines 15-20: Please provide reference for the numbers given.

Page 29, line 6: Please check the use of "cloud burst"

Page 30, line 6: instead of "in danger" please use the more technical term "at risk"

All Maps (figures): please insert a North arrow and a measured grid (geographical coordinates).

Figure 1 would benefit from an additional small overview map on the Black Sea with
the surrounding countries.

Figure 3: what is meant by "line of height mark of the dangerous phenomenon" – needs clarification.

Figure 4: The legend is not logic to me; both points show "number of events (of) no more than 1-2" and ". . .less than 3" – needs clarification.

Figure 5: The legend needs clarification; why the isobar in proximity to the river has higher numbers? Would it be possible just to use "1, 2 and 3 m above water level" or sth. similar?

Figure 6: What is meant by "social important objects"? – infrastructure? residential housing?

Figure 11: What does the small graph stand for? – needs explanation.

Over all, the materials presented are definitely worth being included in the target journal. I suggest that the authors perform a minor revision of the manuscript before it will be published.

―――――――――――――――――

---

## Author Comment (AC1) · 24 Mar 2016

The authors very much appreciated your constructive comments and incorporated all of them into the revised paper. Partly: 1. Abstract, line 13: insert a "the" before Krasnodar Done 2. Abstract, line 16 : the verb is missing in this sentence We changed sentence a little: Assessments of seasonal and maximum flow of the Black Sea Coast Rivers for the period of hydrometric measurements describe regularities of change of the occurrence of inundations and their characteristics on the coastal terrain, within a year and on perennial time scale. 3. Introduction, first and second para: please provide evidence for the numbers given, e.g. references. We inserted references: According to numerous data on the floods in the river mouths and in the coastal zones of European part of Russia (from the 18-20th century until 2013), collected by N. I. Alekseevsky et al. (2013), materials of Taratunin (2000) and other sources 4.

Objectives: : :: please check the use of "km2" (same page 6, lines2 and 3) Yes, we changed -400 km2 5. Objectives: : :, line 14: please provide evidence for the number given, e.g. a reference We inserted references: Magritsky et al., 2013; Data of Federal State Statistics Service in Krasnodar region; Berlin and Petrov, 2015 6. Hydrological data: : :, line 22: replace "for days" by "diurnal". This paragraph describes data which is not represented in Table 1. Could you please add this information to Table 1? Yes, we replaced "for days" by "diurnal" In the tables 1 and 2 runoff data (mean, seasonal, maximum) are presented. These data allow readers to get ideas about water regime, mean and maximum runoff, their statistical parameters. Data, which we used in the mentioned paragraph partly were used in the tables. Unfortunately, synthesis of other data in the form of similar tables, for example, of water levels, is impossible because of numerous violation of uniformity of observation rows due to destruction, transfer, closing of hydrological gauges. The data, which have not been inserted in tables 1 and 2, were attracted by authors for confirmation of cases of floods, for estimation of some characteristics of flooding zones and the rain floods causing them and etc. 7. Inundations: : :, line 16: what is meant by the term "mastered terrains" (same page 11, line 5)? Yes, we put "undeveloped terrains" instead "not mastered terrains" 8. Inundations: : :, line 19/20: should be "Alekseevsky and Magritsky (2013)" yes, done 9. Inundations: : :, page 10, lines 1 and 23: please explain the type 1 inundations a bit more, and take care to use the same wording ("Inundations of mixed type (No1): : :"versus "mixed type 1 inundations" since it makes the text more accessible when using similar expressions. Yes, we changed everywhere "inundations of mixed type (No1)" and added in according paragraph next explanation: Inundations of the mixed type (âĎŰ1) – river-flow and rainfall origin – are next by their occurrence. In general, purely rainfall inundations which also are frequent at the coast are caused by heavy rainfall over the developed areas and by the "inability" of the terrain to quickly absorb or drain rainwater into surface and underground water bodies. The magnitude of rainfall inundations increases if storm drains are functioning badly, therefore in the obvious and dangerous form they happen in settlements, and their frequency increases with

the increase of the area of the urbanized terrains. That is why, they are also named urban inundations. In the basins of mountain and foothill rivers heavy rainfall, first, leads to a rapidly developing flooding by powerful overland streams formed by rainfall waters and by waters of "revived" temporary watercourses. Secondly, rainfalls induce high and fast-moving river floods, which are accompanied by river-flow inundations within the same settlements. The same cause and synchronization of these inundations, the difficulties in division zones of flooding by river and rainfall waters, and the corresponding damages, as well as lack of data, do not allow considering these events separately. They are named inundations of mixed type (No1) 10. Page 13, line 8: please provide a short explanation on the "western jet 2". We replaced "western jet 2"on the planetary frontal zone and added the explanation: Thirdly, activation of planetary frontal zone, the axis of which passes about 40 degree latitudes, leads to intensification of cyclogenesis over the Mediterranean. It is often observed in the autumn, when the active frontogenesis combined with high sea surface temperature. 11. Page 13, lines 19/20: Please check reference. Done: Rasmussen E.A.,, Turner J. Polar lows. Mesoscale Weather Systems in the polar region. // Cambridge Press., 2003 12. Page 15, lines 15-20: Please provide reference for the numbers given. Yes, we added: according to data, collected during field researches 13. Page 29, line 6: Please check the use of "cloud burst" We changed "cloud burst" onto "extreme rainfall" 14. Page 30, line 6: instead of "in danger" please use the more technical term "at risk" Yes, we changed Figures 15. All Maps (figures): please insert a North arrow and a measured grid (geographical coordinates). Done. We added a North arrow and a measured grid (geographical coordinates) 16. Figure 1 would benefit from an additional small overview map on the Black Sea with the surrounding countries Done, we added the additional small overview map on the Black Sea with the surrounding countries 17. Figure 3: what is meant by "line of height mark of the dangerous phenomenon" – needs clarification. We change names of the lines, and added description into legend: UP and DP height of water levels, the excess of which leads to unfavorable (UP) and dangerous (DP) consequences for the population and the

economic activities 18. Figure 4: The legend is not logic to me; both points show "number of events (of) no more than 1-2" and ": : :less than 3" – needs clarification. We changed the legend accordingly 19. Figure 5: The legend needs clarification; why the isobar in proximity to the river has higher numbers? Would it be possible just to use "1, 2 and 3 m above water level" or sth. similar? We clarified the legend – these isolines are water depths 20. Figure 6: What is meant by "social important objects"? – infrastructure? Residential housing? We made more detailed legend, which contains residential, industrial, resort areas separately 21. Figure 11: What does the small graph stand for? – needs explanation. We added explanation, that small graph is the lower part of the curve (zoom).

Please also note the supplement to this comment:
http://www.nat-hazards-earth-syst-sci-discuss.net/nhess-2015-335/nhess-2015-335-AC1-supplement.zip

---

## Referee Comment (RC2) · Anonymous Referee #2 · 29 Mar 2016

("general comments") The work Titled "Causes and systematics of inundations of the Krasnodar territory on the Russian Black Sea coast" is very usefull for scientific and practical purposes.

("technical corrections": typing errors, etc.) The language need to be reviewed, some suggestions were made on the text enclosed.

("specific comments") Distincts items of contents are not well presented, i.e. " 2 Objectives of research" where the authors only did the decription of the area. . ..

The organization of all content in the classic way " 1.° Introduction 2.° Material and Method 3.° Results and discussions 4.° Conclusions " will be better.

Please also note the supplement to this comment:

[Figure]

http://www.nat-hazards-earth-syst-sci-discuss.net/nhess-2015-335/nhess-2015-335-RC2-supplement.pdf

[Figure]

**Supplement:**

[Figure]

[Figure]

[revised manuscript text omitted]
≈0.79 km$^2$), Mezyb (settlement Divnomorskiy; F≈0.44 km$^2$), Shapsukho (settlement Lermontovo; F≈0.83 km$^2$), Nechepsukho (settlement Novomikhailovskiy), Tuapse (Tuapse; F≈2.08 km$^2$), Vulan (settlement Arkhipo-Osipovka; F≈0.73 km$^2$), and Tu (settlement Olginka) during the big and catastrophic inundations.

[Figure]

[Figure]

a) Letters or numbers should be used, only one, but the same...

Figure 7. Consequences of a catastrophic high water in August 2012. Settlement Novomikhaylovskiy (Tuapse district). a -depositions of river sediments and refuse at in the city stadium (August, 2012; www.livekuban.ru); b -clearing the channel from deposits and vegetation (October, 2012, D.V.Magritsky); c - «turbid plume» in the mouth of the Nechepsukho river (August, 2012; http://ria.ru/)

[Figure]

[Figure]

Figure 8. Interannual changes of the number of inundations at the Black Sea coast of Krasnodar territory (a) and fluctuation of total duration for a year of groups of types of atmospheric circulation of Northern hemisphere in B.L.Dzerdzeevsky's typification in 1945-2013 (b; by data from site: http://atmospheric-circulation.ru/datas/).

[Figure]

[Figure]

[Figure]

Figure 9. Long-term fluctuations of the maximum water discharges of the rivers Vulan (a), Tuapse (b), Shakhe (c) and Sochi (d): 1 – maximum water discharges, 2 –mean annual water discharge, 3 – maximum deviation from the mean annual water discharge by 1 σ.

[Figure]

[Figure]

[Figure]

a)          b)

Figure 10. Mouth of the Dagomys River (the city-resort of Sochi) in the beginning of the 20th century and in the beginning of the 21st century. The left picture (a) – S.M. Prokudin-Gorskiy's photo (1910-

1915, Library of the Congress of the USA). The right picture (b) – a photo from a resource http://www.panoramio.com –user №6172839l (on August 18th, 2013)

[Figure]

[Figure]

Figure 11. Empirical relationship between direct material damage (millions of US dollars) and type of inundation (river-flow and mixed type 1), or probability of $Q_{max}$

[Figure]

Figure 12. An example of numerical modelling of inundation in the valley of the Ashamba River (terrain of city-resort Gelendzhik): a) terrain before flood, b) calculated maximum flood depths

---

## Author Comment (AC2) · 22 Apr 2016

The authors greatly appreciate constructive comments of Anonymous Reviewer #2 that helped us to better focus and improve the paper. Regarding specific comment 1 " Distinct items of contents are not well presented, i.e. " 2 Objectives of research" where the authors only did the description of the area". We are very sorry for the translation and technical mistakes. This part initially was named "Objects of research", now we changed it into "Study area". Objectives of research are described in the abstract and the introduction parts and focus on the analysis of inundation situations on the Black Sea coast of the Krasnodar territory for the period from 1945 until 2013, definition of the main types, the analysis of synoptic factors of the formation of extreme rainfalls and rainfall floods, the analysis of the efficiency of the measures applied at the coast to mitigate inundations and their after-effects.

[Figure]

Regarding specific comment 2 "The organization of all content in the classic way" 1._ Introduction 2._ Material and Method 3._ Results and discussions 4._ Conclusions " will be better." First part of the paper is organized in a classic way 1. Introduction, 2. Study area, 3. Hydrological data and methods of research. We provided separately section 4 to describe our classifications of inundations. Further results and discussions are then grouped into several sections: 5. Synoptic conditions of the formation of high floods, 6. Features and regularities of flood routing, 7. Temporal regularities of inundations 8. Geographical features and hazards of inundations 9. Countermeasures for inundations and their efficiency. 10. Conclusions.

Such division intentionally can better help to pick out different aspects concerning causes and systematics of inundations of the Krasnodar territory on the Russian Black Sea coast.

All technical corrections and suggestions regarding language improvement are incorporated into revised paper (see supplement file, the changes marked blue).

Please also note the supplement to this comment:
http://www.nat-hazards-earth-syst-sci-discuss.net/nhess-2015-335/nhess-2015-335-AC2-supplement.zip